# EVLP: Learning Unified Embodied Vision-Language Planner with Reinforced Supervised Fine-Tuning

**Xinyan Cai**[1,2] **Qiang Guan**[1,2*] **Shiguang Wu**[3†] **Dafeng Chi**[3] **Yuzheng Zhuang**[3]
**Jianye Hao**[3] **Xingyue Quan**[3]

[1]Institute of Automation, Chinese Academy of Sciences (CASIA)
[2]School of Artificial Intelligence, University of Chinese Academy of Sciences
[3]Huawei Noah's Ark Lab

## Abstract

In complex embodied long-horizon manipulation tasks, effective task decomposition and execution require synergistic integration of textual logical reasoning and visual-spatial imagination to ensure efficient and accurate operation. Current methods fail to adopt a unified generation framework for multimodal planning, lead to inconsistent in multimodal planning. To address this challenge, we present **EVLP (Embodied Vision-Language Planner)**, an innovative multimodal unified generation framework that jointly models linguistic reasoning and visual generation. Our approach achieves multimodal planning for long-horizon tasks through a novel training pipeline incorporating dynamic pretraining and reinforced alignment. Our core innovations consist of three key components: **1) Unified Multimodal Generation Framework**: For understanding, We integrate semantic information with spatial features to provide comprehensive visual perception. For generation, we directly learn the joint distribution of discrete images for one-step visual synthesis, enabling coordinated language-visual modeling through learnable cross-modal attention mechanisms. **2) Dynamic Perception Pretraining**: We propose a bidirectional dynamic alignment strategy employing inverse dynamics tasks and forward dynamics tasks, effectively strengthening multimodal correlations within a unified feature space. **3) Reinforced Supervised Fine-Tuning**: While conducting instruction-based fine-tuning in the unified generation space, we construct a reinforce loss to align the spatial logic between textual actions and generated images, enabling the model to acquire spatio-aware multimodal planning capabilities. Comprehensive evaluations on multiple complex tasks demonstrate that EVLP significantly outperforms competitive baselines in both instruction execution accuracy and task success rate, benefiting from its unified multimodal architecture and well-designed training pipeline. Extensive ablation studies further validate the rationality of our framework design.

## 1 Introduction

In the realm of long-term planning for embodied intelligence, current task decomposition methodologies predominantly adopt two distinct technical paradigms: **language planning** parses high-level instructions (e.g., "tidy up the room") into sequential atomic actions (e.g., "pick up clothes → place them in the closet") to explicitly specify **what to do** (What) (Ahn et al., 2022; Mu et al., 2023a; Guo et al., 2024; Zhang et al., 2025b; Zhao et al., 2025), while **visual planning** resolves **how to accomplish it** (How) through generating intermediate visual representations (e.g., depicting "clothes inside the closet") (Black et al., 2023b; Du et al., 2023; Soni et al., 2025). In contrast, emerging multimodal planning frameworks (Ni et al., 2024b) synergistically produce linguistic action sequences and corresponding visual targets, thereby bridging the critical gap between procedural execution and goal

---

*✉ Corresponding author: qiang.guan@ia.ac.cn
†✉ Corresponding author: wushiguang@huawei.com

achievement. This dual-representation approach demonstrates significant potential for overcoming existing planning limitations in embodied systems (Black et al., 2023b; Ni et al., 2024b), positioning them as a pivotal research direction for next-generation intelligent agents.

Recent advances in unified multimodal generative models (Ge et al., 2023b;a; Team, 2025) have demonstrated unprecedented cross-modal generation capabilities, differing from traditional multi-modal understanding models (Liu et al., 2023a;b; 2024; Li et al., 2022) or architectures that generate through external diffusion (Ge et al., 2023c). These architectures unify text and image generation within a single Transformer backbone, showcasing unified understanding and generation capabilities across different modalities (Ma et al., 2025; Zhou et al., 2024; Shi et al., 2025; Wu et al., 2025b). Their inherent scalability (Wu et al., 2024; Chen et al., 2025) and emergent cross-modal synergy (Xie et al., 2024) enable mutual reinforcement between text-image comprehension and synthesis tasks. This naturally raises an important research question: *Can such unified multimodal architectures be leveraged to construct Vision-Language planners capable of resolving robotic manipulation tasks that require complex multi-step instruction decomposition?*

Developing a unified multimodal planner presents three key technical hurdles. First, while existing multimodal models excel at matching images with text descriptions (*e.g.*, generating a cat image from "a cat on a sofa"), embodied planning requires deeper understanding. The model must not only recognize objects in images (*e.g.*, "cup on table") but also precisely grasp where objects are located in space—critical for planning physical actions like grasping or moving. Second, traditional multimodal tasks (*e.g.*, image captioning or visual QA) focus on static understanding. Embodied planning adds a temporal dimension: successful models must track how actions transform the environment. Imagine predicting how "pouring water" changes a scene from (1) cup upright to (2) cup tilted—this demands reasoning about state transitions rather than just recognizing static patterns. Third, conventional maximum likelihood training treats all visual details equally. While this works for generating photorealistic images, operational tasks prioritize functional consistency over visual perfection. For example, when planning "put clothes in closet", we care more about the clothes' final position (in closet) than their exact folds or shading. Current training paradigms lack mechanisms to prioritize such task-relevant aspects over irrelevant visual details.

To address this challenge, we present **EVLP (Embodied Vision-Language Planner)** , a novel framework that seamlessly unifies language reasoning and visual imagination within a single multi-modal architecture for long-term manipulation tasks. EVLP innovatively integrates: 1) A dual-tower vision module that decouples understanding and generation—employing SigLIP's semantic encoder enhanced by trainable detail compensators to mitigate systematic visual blind spots, paired with discrete tokenization via MAGVIT2 (Yu et al., 2024) for direct image synthesis; 2) A bidirectional dynamics-aware pre-training paradigm that equips the model with coherent reasoning-imagining capabilities by learning from forward/inverse dynamic prediction tasks; 3) Reinforced Supervised Fine-Tuning (RSFT), which uses maximum likelihood to supervise the entire token distribution while dynamically reinforcing spatial consistency between language actions and generated images through policy gradients. Compared to existing methods, EVLP demonstrates significant improvements in instruction following and task success rates in complex manipulation tasks, establishing a promising and inspiring paradigm for long-term operations. We also conducted extensive ablation experiments to provide references for future research.

## 2 METHOD

EVLP leverages pre-trained LLMs through a unified transformer architecture for multimodal generation, processing both stepwise linguistic instructions and visual subgoal images, which providing linguistic milestones and perceptual anchors to guide action sequences in long-horizon tasks. Our technical exposition proceeds systematically: Section 2.1 details the multimodal architecture enabling unified generation; Section 2.2 examines dynamic perception pre-training for learning environmental dynamics; and Section 2.3 introduces reinforced supervised fine-tuning phase, which enabling the model to acquire spatio-aware multimodal planning capabilities. Our overall framwork is illustrated in Figure 1.

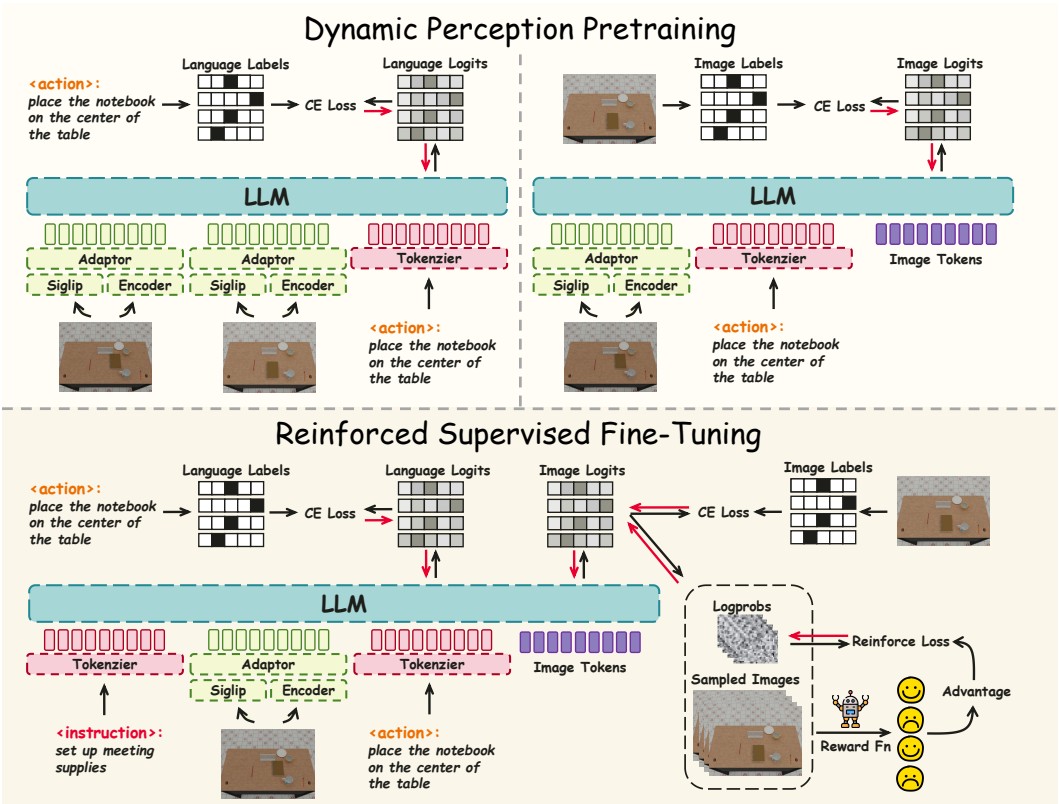

Figure 1: Our overall framework diagram. In terms of the model architecture, we adopt a vision tower design that integrates understanding and generation. For image understanding, we combine SigLIP with a learnable spatial encoder, while for image generation, we introduce image tokens to achieve one-step generation. Regarding the training pipeline, we design a two-stage framework: dynamic perception pretraining (illustrated above) and reinforced supervised fine-tuning (illustrated below). The black arrows represent the forward process, while the red arrows indicate the backward process.

## 2.1 UNIFIED MODEL FOR MULTIMODAL GENERATION

Modern unified multimodal generative models predominantly adopt two architectural paradigms: diffusion-based and autoregressive approaches. Diffusion architectures (Ho et al., 2020; Peebles & Xie, 2023; Xie et al., 2024) aim to adapt large language models (LLMs) for image denoising through multi-step iterative refinement (Zhou et al., 2024), while autoregressive methods (Liu et al., 2025; Team, 2025; Qu et al., 2024; Yu et al., 2024; Wu et al., 2025a) employ unified tokenization strategies for both modalities under a single learning objective (Wu et al., 2024; 2025b).

We identify critical limitations in both frameworks: the diffusion objective exhibits fundamental discrepancy with LLMs' pre-training tasks (Wu et al., 2025a), creating optimization conflicts during multimodal adaptation, while autoregressive modeling imposes artificial sequential dependency on inherently non-sequential visual data. Notably, when applying reinforcement learning for preference optimization (von Werra et al., 2020; Hu et al., 2024; Fan et al., 2023; Black et al., 2023a), both architectures require computationally prohibitive multi-step sampling to generate independent candidates - a bottleneck that scales poorly with model size.

Our solution introduces a lightweight multimodal architecture that seamlessly integrates multimodal capabilities into existing LLM frameworks. The key innovation lies in a sampling-efficient generator that produces multiple independent samples through **single-forward propagation**, enabling joint optimization of supervised and reinforcement learning objectives during fine-tuning as detailed in Section 2.3.

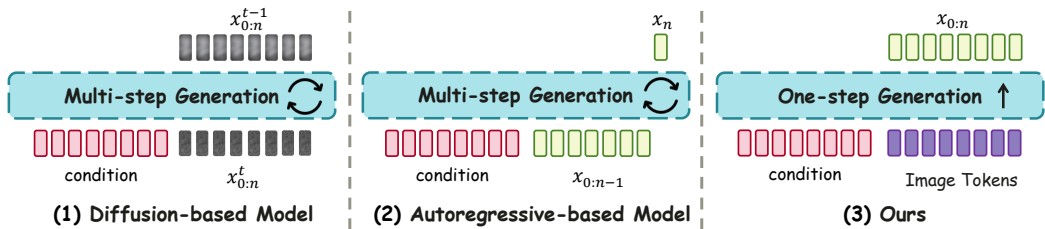

Figure 2: (1) **Diffusion-based Model** formulates image generation as $x_{0:N}^{t-1} \sim p(\cdot|c, x_{0:N}^t)$. When sampling $n$ samples from distribution $p(\cdot|c)$, the model requires $n \times T$ forward passes, where $T$ denotes the diffusion denoising steps. (2) **Autoregressive-based Model** formulates image generation as $x_{0:N}^{t-1} \sim p(\cdot|c, x_{0:N}^t)$. When sampling $n$ samples from distribution $p(\cdot|c)$, the model requires $n \times N$ forward passes, where $N$ represents the token count. (3) **Our Model** directly models $p(\cdot|c)$, enabling the sampling of $n$ samples with only ***one forward pass***.

**Vision Tower**    To enable large language models to understand and generate images, we developed our Vision Tower. For **image understanding**, we utilize Siglip to extract high-level semantic information from images. Additionally, to address the spatial detail information that Siglip may overlook (Tong et al., 2024b), we incorporated a low-level visual encoder pre-trained with an image reconstruction loss, focusing on the intricacies within the images. During training, we keep Siglip's weights frozen while allowing the low-level visual encoder to participate in the process. Our goal is for both components to focus on different levels of visual signals, which are then fed into the LLM through an adapter. For **image generation**, to seamlessly integrate with the existing training objectives of the LLM, we trained a lookup-free quantizer within the Open-MAGVIT2 framework (Luo et al., 2025). This quantizer maintains a codebook of size K = 262,144 and encodes images with a resolution of 256×256 into 16×16 discrete tokens.

**Generation Architecture**    To build a unified modality model, we combine the Vision Tower with a pretrained large language model (LLM). For images generation, we adopt a one-step generation approach: as shown in Figure 2. We assume that images can be processed as $N$ discrete or continuous tokens. The Diffusion-based Model and Autoregressive-based Model model the generation process as $x_{0:N}^{t-1} \sim p(\cdot|c, x_{0:N}^t)$ and $x_N \sim p(\cdot|c, x_{0:n-1})$ respectively. To obtain the complete conditional distribution $x_{0:N} \sim p(\cdot|c)$, the model needs to perform multiple autoregressive forward passes. In this paper, we take a direct approach: we allow the LLM to model $x_{0:N} \sim p(\cdot|c)$ directly. Practically, we introduce a set of learnable image tokens, which are input into the transformer along with the conditions, and use the image tokens to directly predict the discrete token IDs of the image. This modeling approach avoids introducing unnecessary priors and seamlessly integrates with the original learning objectives of the LLM. By integrating the Vision Tower with the LLM, we have developed a unified multimodal generation model.

## 2.2 DYNAMIC PERCEPTION PRETRAINING

To enable the model to learn the understanding and generation of multimodal interactions, we designed a dynamic perception pre-training phase. For embodied agents, understanding the environment is crucial. This not only requires agents to comprehend images and text individually but also to grasp the interactions between them. The model must understand the differences between two distinct states and be able to infer how taking a specific action will alter the state.

Based on these considerations, we collected a dataset $\mathcal{D} = \{\mathcal{T}_0, \mathcal{T}_1, \ldots, \mathcal{T}_l\}$ with transitions defined as $\mathcal{T} = \{x_t, a_t, x_{t+1}\}$, where $x$ represents image observations and $a$ denotes language actions. Building on this foundation, we designed a pre-training phase based on Inverse Dynamic Learning and Forward Dynamic Learning.

**Inverse Dynamic Task**    To train the model's perceptual capabilities while maintaining its text generation abilities, we designed the Inverse Dynamic task. Specifically, given the dataset $\mathcal{D}$ and a prompt $\mathcal{P}$ such as: *"What is the action between  <ImageHere> </Img> and  <ImageHere> </Img>? Please infer the actions that took place"*, the objective function is the conditional log-likelihood of the action token sequence:

$$\mathcal{L}_{\text{Inverse Dynamic}} = -\mathbb{E}_{(x_t, a_t, x_{t+1}) \sim \mathcal{D}} \left[ \frac{1}{L} \sum_{i=1}^{L} \log P(a_t^{(i)} \mid a_t^{(<i)}, x_t, x_{t+1}; \theta) \right], \qquad (1)$$

where $L$ represents the token length of the action sequence $a_t$, $a_t^{(i)}$ denotes the $i$-th token in the action sequence, $a_t^{(<i)}$ refers to all historical tokens before position $i$, and $\theta$ represents the model's trainable parameters. By optimizing the Inverse Dynamic Loss $\mathcal{L}_{\text{Inverse Dynamic}}$, we enhance the model's image understanding and dynamic comprehension abilities while maintaining its text generation capabilities.

**Forward Dynamic Task**    To enable the model to learn image generation and understanding while enhancing its reasoning capabilities, we designed the Forward Dynamic task. The model needs to infer the next observation $x_{t+1}$ based on the current observation $x_t$ and the language action $a_t$. Specifically, given the dataset $\mathcal{D}$ and a prompt $\mathcal{P}$ such as: *"What will happen if  <ImageHere> </Img> takes the action like <ActionHere>? Please generate an image of the next state"*, the objective function is the conditional log-likelihood of the image token sequence:

$$\mathcal{L}_{\text{Forward Dynamic}} = -\mathbb{E}_{(x_t, a_t, x_{t+1}) \sim \mathcal{D}} \left[ \log P(x_{t+1}^{(0:N)} \mid x_t, a_t; \theta) \right], \qquad (2)$$

where $N$ represents the token length of the image sequence $x_t$, $a_t$ denotes the action sequence, and $\theta$ represents the model's trainable parameters. By optimizing the Forward Dynamic Loss $\mathcal{L}_{\text{Forward Dynamic}}$, we enhance the model's ability for image generation and understanding while improving its dynamic reasoning capabilities.

**Joint Pretraining**    During the pre-training phase, we employed a dual-task curriculum where inverse dynamics modeling and forward dynamics prediction were co-trained on identical multimodal datasets. This co-optimization strategy establishes a unified framework for processing multimodal inputs and generating coordinated outputs, effectively forming the foundation for the model's emergent world modeling capabilities through integrated cross-modal reasoning.

## 2.3    REINFORCED SUPERVISED FINE-TUNING

After pre-training, the model has acquired the ability to unify the processing of visual and textual information, along with a preliminary dynamic understanding capability. Next, we need to leverage the model's perception and reasoning abilities to construct a multimodal planning agent. We require the model not only to understand the objective environmental transition functions but also to make proactive inferences based on given goals, breaking down complex tasks into simpler actions. Based on these considerations, we collected a dataset $\mathcal{D} = \{\mathcal{T}_0, \mathcal{T}_1, \ldots, \mathcal{T}_l\}$ containing high-level instructions and low-level language actions, where the basic elements are $\mathcal{T} = \{g, x_t, a_t, x_{t+1}\}$, with $g$ representing the high-level instruction, $x$ denoting the image observation, and $a$ indicating the language action. On this basis, we designed a Reinforced Supervised Fine-Tuning(RSFT) phase.

**Maximum Likelihood for Vision-Language Planning**    To learn the model's perception and reasoning abilities while applying our multimodal framework for vision-language joint planning, we first perform supervised fine-tuning (SFT) on the model. Specifically, the model needs to infer the appropriate language action $a_t$ based on the given high-level instruction and the current observation $x_t$, as well as the observation $x_{t+1}$ that results from taking action $a_t$. Given the dataset $\mathcal{D}$ and a prompt $\mathcal{P}$ such as: *"You are given the current image observation  <ImageHere> </Img> and the given task instruction: <TaskHere>. Please make the next step action decision and generate the next state image"*, the objective function is the joint conditional log-likelihood of the action tokens and image tokens:

$$\mathcal{L}_{\text{SFT}} = -\mathbb{E}_{(g, x_t, a_t, x_{t+1}) \sim \mathcal{D}} \left[ \frac{1}{L} \sum_{i=1}^{L} \log P(a_t^{(i)} \mid a_t^{(<i)}, g, x_t; \theta) + \log P(x_{t+1}^{(0:N)} \mid g, x_t, a_t^{0:L}; \theta) \right],$$
$$(3)$$

where $L$ represents the token length of the action sequence $a_t$, $a_t^{(i)}$ denotes the $i$-th token in the action sequence, $a_t^{(<i)}$ indicates all historical tokens before position $i$, $N$ is the token length of the image sequence $x_t$, and $\theta$ are the model's trainable parameters. By optimizing the joint conditional log-likelihood $\mathcal{L}_{\text{SFT}}$, the model learns multimodal reasoning capabilities.

**Policy Gradient for Dynamic Alignment**  The supervised objective $\mathcal{L}_{\text{SFT}}$ minimizes cross-entropy between the model's conditional joint distribution $p_\theta(x_{0:N}|c)$ and target data distribution. While effective for static pattern alignment, maximum likelihood estimation suffers from two critical limitations in dynamic reasoning tasks: (1) *perceptual over-specification* - enforcing unnecessary constraints on task-irrelevant details (e.g., table texture consistency), and (2) *causal under-constraint* - failing to model physical dynamics that govern state transitions. Reinforcement learning (Sutton & Barto, 1998) addresses this mismatch through reward-weighted policy gradients, where human preferences guide the learning of physically plausible trajectories. Our key innovation seamlessly integrates this paradigm by exploiting the model's single-forward multi-sample generation capability to compute advantage-weighted gradients without additional computational overhead.

Since our model directly models $x_{0:N} \sim p(\cdot|c)$, we can independently sample multiple samples in a single forward pass. Specifically, we designed a Dynamic Alignment reward function $r = R(x)$, used to measure whether the dynamics of the generated image are consistent with the real dynamics, the specific definition of the reward function can be found in the Appendix C.1. During the forward pass, we sample $K$ samples $x^k \sim P(x_{t+1}^{(0:N)} \mid g, x_t, a_t^{0:L})$ and obtain the reward for each sample using the reward function. We normalize these rewards within each batch to calculate the advantage for each sample and optimize with the equation 4.

By explicitly encoding the dynamic alignment prior through the reward function $R(\cdot)$, we strengthen the model's dynamic alignment capabilities. However, due to the use of high-variance gradient estimators, reinforcement learning exhibits instability during training, which can lead to training collapse. Therefore, we combine maximum likelihood with policy gradients to achieve stable joint optimization.

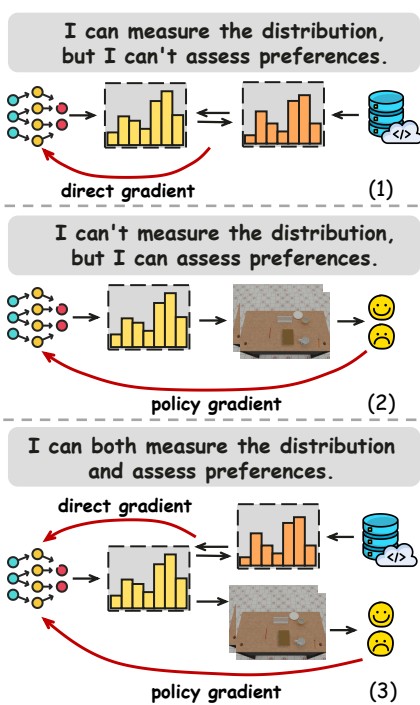

Figure 3: (1) SFT optimize model by minimizing KL divergence between model outputs and dataset distributions. It lacks per-sample preference alignment. (2) Reinforcement Learning (RL) aligns preferences through per-sample feedback, focusing on reward maximization but risking distribution shifts. (3) Reinforcement Supervised Fine-Tuning (RSFT) combines distribution constraints with sample optimization, enforcing preference alignment under maximum likelihood constraints.

$$\mathcal{L}_{\text{RL}} = -\mathbb{E}_{(g,x_t,a_t) \sim \mathcal{D}, x_{t+1}^k \sim P(\cdot|g,x_t,a_t;\theta)} \left[ \frac{1}{K} \sum_{k=1}^{K} A_k \cdot \log P(x_{t+1}^k \mid g, x_t, a_t^{0:L}; \theta) \right]. \quad (4)$$

**Joint optimization**  In practice, we constrain the model's overall distribution using maximum likelihood and leverage policy gradient to enhance dynamic alignment capabilities (Figure 3). Specifically, we optimize our model with the following objective:

$$\mathcal{L} = -\mathbb{E}_{(g,x_t,a_t,x_{t+1}) \sim \mathcal{D}} \left[ \mathcal{L}_{\text{SFT}} + \lambda \cdot \mathcal{L}_{\text{RL}} \right]. \quad (5)$$

The first term enforces global alignment between language and vision, while the second improves dynamic consistency via preference-aware sampling. Together, they enable physically plausible action

Table 1: The evaluation of success rate between baselines and we report the mean and variance across 5 seeds.

| Model | Blocks | | | Letters | | |
|---|---|---|---|---|---|---|
| | **Stacking** | **Sort** | **Matching** | **Shape** | **Orders** | **Spell** |
| CLIPort | $18.4 \pm 3.2$ | $19.2 \pm 4.6$ | $17.8 \pm 2.9$ | $9.8 \pm 1.4$ | $8.1 \pm 2.7$ | $2.3 \pm 0.8$ |
| PAR | $34.7 \pm 5.5$ | $32.8 \pm 6.3$ | $31.1 \pm 4.4$ | $31.5 \pm 5.8$ | $30.7 \pm 4.9$ | $27.3 \pm 7.2$ |
| EmbodiedGPT | $48.6 \pm 6.7$ | $49.1 \pm 5.9$ | $43.4 \pm 7.8$ | $40.9 \pm 6.4$ | $48.2 \pm 7.5$ | $52.7 \pm 6.2$ |
| SuSIE | $34.1 \pm 3.8$ | $32.6 \pm 4.1$ | $33.2 \pm 5.7$ | $37.8 \pm 6.6$ | $35.2 \pm 4.3$ | $34.1 \pm 7.4$ |
| CoTDiffusion | $47.9 \pm 6.0$ | $44.3 \pm 7.6$ | $56.6 \pm 5.2$ | $46.1 \pm 6.5$ | $53.9 \pm 4.8$ | $44.8 \pm 7.9$ |
| PERIA | $63.9 \pm 5.8$ | $65.0 \pm 6.4$ | $72.3 \pm 7.1$ | $60.6 \pm 5.2$ | $65.2 \pm 6.7$ | $71.1 \pm 7.5$ |
| EVLP (ours) | $\mathbf{79.4 \pm 7.9}$ | $\mathbf{77.3 \pm 4.3}$ | $\mathbf{82.5 \pm 6.1}$ | $\mathbf{75.3 \pm 4.4}$ | $\mathbf{78.2 \pm 7.3}$ | $\mathbf{81.8 \pm 6.5}$ |

sequences and subgoal-conditioned images from high-level instructions, supporting autonomous planning under real dynamics. Implementation details and pseudocode are provided in Appendix C.3.

## 3 EXPERIMENTS

### 3.1 EXPERIMENT SETUP

**Benchmark & Settings**   We evaluate across long-horizon manipulation environments, configuring the model to generate language-guided actions and visual subgoals from high-level instructions, with a low-level policy executing the plans (implementation details in Appendix G). Experiments use **LoHoRavens** (Zhang et al., 2023), a Ravens-based benchmark where we consider 11 language-conditioned *block* tasks and extend with 9 challenging *letter* tasks, and **Meeting Preparation**, an in-house simulator for arranging desktop objects under varied backgrounds, table types, and object categories, enabling deeper analysis of planning in realistic, diverse settings.

**Baselines**   We compare against four planning paradigms: (i) **end-to-end** imitation learning with *CLIPort* (Shridhar et al., 2022), which directly maps high-level instructions to actions without an explicit planner; (ii) **language planning**, including *PAR* (Zhang et al., 2023), which uses a VLM reporter and an LLM planner, and *EmbodiedGPT* (Mu et al., 2023b), which replaces the LLM+VLM stack with a stronger MLLM after instruction tuning; (iii) **vision planning**, represented by *SuSIE* (Black et al., 2023b), which edits images to form subgoals for simple steps, and *CoTDiffusion* (Ni et al., 2024a), which introduces a semantic alignment module for chain-of-thought subgoal generation; and (iv) **multimodal planning** with *PERIA* (Ni et al., 2024b), which jointly plans language actions and image conditions with an LLM while a diffusion model renders visual subgoals.

### 3.2 MAIN QUANTITATIVE RESULTS OF SUCCESS RATE

The results on LoHoRavens are shown in Table 1. As expected, end-to-end learning methods performed the worst, struggling to complete tasks due to a lack of intermediate guidance. In contrast, the language planning paradigm explicitly decomposes tasks into stepwise instructions and employs a hierarchical framework consisting of a language planner and a language-conditioned policy, demonstrating greater potential and clearly outperforming the end-to-end approach. Meanwhile, EmbodiedGPT achieved stronger performance through fine-tuning.

The visual planning paradigm generates intermediate keyframes. For example, SuSIE uses an image editing model to directly generate subgoals, while CoTDiffusion produces more refined images through Chain of Thought (CoT) reasoning, enhancing the performance of visual planning. However, CoTDiffusion does not explicitly reason about the instructions, which may lead to semantic inconsistencies in the generated subgoal images. In comparison, PERIA further enhances performance through multimodal planning. This method provides rich subgoal information by jointly planning with language and vision. However, PERIA is limited by its visual perception capabilities and the interaction between modalities, which may result in inaccuracies in the generated target images.

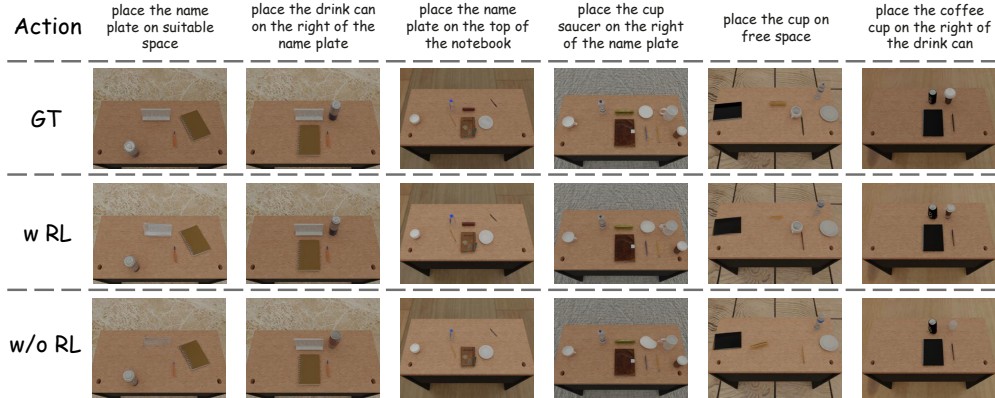

Figure 4: Comparison of generation effects between RSFT and SFT shows that RSFT generates more finely detailed results with better dynamic consistency.

In contrast, our EVLP algorithm integrates the processing of textual instructions and image observations, introducing richer visual perception and designing a framework for unified interaction between text and images. Additionally, we developed a dynamic alignment reward function to optimize the model's dynamic reasoning capabilities through reinforcement learning. Our approach surpasses the baselines across all tasks, achieving optimal performance.

## 3.3 FURTHER ANALYSIS

This section trained multiple variants of EVLP and conducted two sets of experiments on the Meeting Preparation task: first, given instructions and the observation, we tested the model's planning capabilities by measuring the success rate (SR), language accuracy (LA), and LPIPS, SSIM on test set(Table 2). We also tested the image generation capabilities by providing the observation along with actions, asking the model to generate the next state, and evaluated the LPIPS and SSIM(Table 3).

**Ablation on Vision Tower**  To validate the effectiveness of our vision tower, we created two variants: one removing the spatial encoder from the vision tower (EVLP w/o En) and the other excluding Siglip from the architecture (EVLP w/o Se). We first compared their generation performance (Exps. A, B, and C in Table 3). It is evident that the image generation capability of EVLP w/o En significantly declines, as relying solely on Siglip information lacks crucial details regarding positional context. Meanwhile,

Table 2: Comparative analysis of task planning performance across different variants.

| | Method | SR ↑ | LA ↑ | LPIPS ↓ | SSIM ↑ |
|---|---|---|---|---|---|
| A | EVLP | 67.6 | 87.0 | 0.051 | 0.95 |
| B | - w/o En | 56.5 | 82.9 | 0.092 | 0.92 |
| C | - w/o Se | 50.1 | 73.9 | 0.116 | 0.89 |
| D | - w/o IDM | 63.9 | 83.6 | 0.052 | 0.95 |
| E | - w/o FDM | 26.8 | 72.1 | 0.192 | 0.84 |
| F | - w/o RL | 62.2 | 87.4 | 0.054 | 0.95 |
| G | - RL only | 0.0 | 14.0 | 0.712 | 0.29 |

although EVLP w/o Se shows less degradation in image generation tasks, it becomes clear in specific multimodal reasoning tasks (Exps. A, B, and C in Table 2) that the absence of Siglip significantly reduces the model's language planning abilities, leading to suboptimal overall performance. This is due to the spatial encoder alone lacking essential semantic information, making it challenging for the LLM to process. In contrast, the combination of both components achieves the best results.

**Ablation on Generation Architecture**  To test if one-step generation(Fig. 2) improves quality, we build EVLP-AR, which keeps the same components but generate autoregressively, following (Wu et al., 2024). Comparing EVLP with EVLP-AR (Exps. A,D in Tab. 3) shows the AR variant markedly degrades fidelity and increases hallucinations. We believe this is primarily due to two reasons: (i) imposing an unnatural causal-sequence prior on images and (ii) cumulative AR errors that hinder precise operations.

Table 3: Comparative analysis of image generation performance across different variants.

| | Method | LPIPS ↓ | SSIM ↑ |
|---|---|---|---|
| A | EVLP | 0.046 | 0.95 |
| B | - w/o En | 0.087 | 0.87 |
| C | - w/o Se | 0.074 | 0.91 |
| D | - AR | 0.197 | 0.84 |

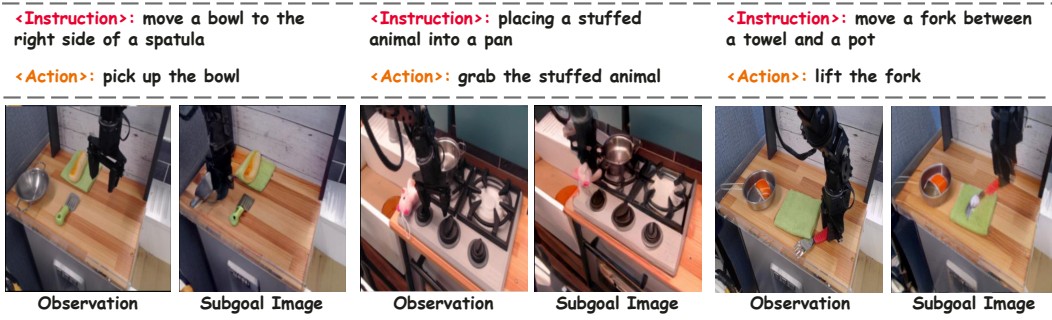

Figure 5: Visualization of Real-World Dataset experiments, showcasing EVLP's planning quality in complex, real-world scenes.

**Ablation on Pretraining** We further disentangle the contributions of two pretraining objectives—IDM for predicting action commands and FDM for predicting state transitions. Using the same architecture, we compare three settings: joint dual-task pretraining (IDM+FDM), w/o FDM and w/o IDM (Exps. A,D,E in Tab. 2). Results show that the joint model performs best overall, underscoring the necessity of combined supervision. Ablations indicate that removing IDM weakens language planning, while removing FDM markedly degrades multimodal planning. Together, these findings highlight the pivotal role of bidirectional pretraining in modality alignment.

**Ablation on RSFT** In Section 2.3, we introduced the Reinforcement Supervised Fine-Tuning (RSFT) framework, which optimizes both likelihood and dynamic rewards. Comparative analysis (Exps. A, F, and G in Table 2) of RSFT (learning by Equation 5), traditional supervised fine-tuning (SFT) (learning by Equation 3), and RL-only (learning by Equation 4) shows that RSFT performs similarly to SFT on standard image metrics but significantly improves task success rates. Notably, the RL-only baseline exhibits catastrophic policy collapse without SFT regularization, underscoring the necessity of dynamic consistency constraints for stable learning – a key advantage preserved by our hybrid approach. Reward curves on the test set (Figure 6) further confirm our algorithm's effectiveness in reward accumulation compared to traditional SFT methods. The actual generated results (Figure 4), demonstrating that RSFT perform better in dynamic consistency.

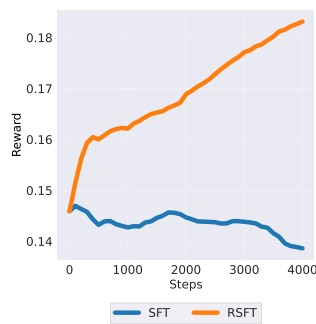

Figure 6: The reward curve on test set during training.

**Real-World Dataset Evaluation** We constructed a real-world robotic manipulation dataset based on the BridgeData v2. This dataset features diverse household tasks (e.g., object grasping, stacking) with recorded robotic arm observations and natural language instructions. We evaluated EVLP's multimodal planning capabilities using LA and LPIPS. As summarized in Table 4, EVLP achieves the best performance across both modalities. In language planning, EVLP obtains a LA of 0.78, surpassing EmbodiedGPT and PERIA, indicating stronger alignment with task intent. In visual planning, EVLP reduces LPIPS to 0.11, compared with 0.23 for SuSIE and 0.17 for PERIA, demonstrating more spatially consistent visual subgoals.

Table 4: Comparative analysis in Real-World Dataset.

| | Method | LA↑ | LPIPS↓ |
|---|---|---|---|
| A | EVLP | 0.78 | 0.11 |
| B | PERIA | 0.75 | 0.17 |
| C | SuSIE | – | 0.23 |
| D | EmbodiedGPT | 0.68 | – |

## 4 CONCLUSION

We introduced **EVLP (Embodied Vision-Language Planner)**, a unified multimodal generation framework that integrates language reasoning and visual imagination for long-horizon manipulation tasks. Through dynamic perception pre-training, EVLP learns spatial relations and environmental dynamics, while our reinforced supervised fine-tuning (RSFT) enables dynamically consistent multimodal planning. Extensive experiments on challenging benchmarks show that EVLP achieves

superior instruction-following accuracy and task success rates compared to strong baselines. Additionally, we conducted detailed analyses to validate the effectiveness of our design. We believe that EVLP highlights the potential of holistic language and vision planning, and we hope this novel paradigm can provide valuable insights for robotics research in long-horizon tasks with complex instructions, moving toward more open embodied scenarios.

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

## A  Related Work

### A.1  Unified MLLM for Multimodal Generation

The success of large language models (LLMs) has spurred growing interest in harnessing them for image generation, leading to a variety of architectural approaches. One straightforward approach is integrating LLMs with image generation models, where the LLM produces conditions to guide the generation process (Ge et al., 2023c; Tong et al., 2024a; Huang et al., 2025). However, these models require additional generative models, which is not conducive to unified multimodal representation and model scaling. To address this, many works have attempted to use a single model for multimodal understanding and generation. Some have explored combining diffusion models with LLMs (Xie et al., 2024; Zhou et al., 2024). Others have focused on discretizing images and using a unified autoregressive objective to model both text and images. Within this area, some works have concentrated on image discretization methods (Wu et al., 2025b;a; Qu et al., 2024), while others have focused on the integration of multimodal tokens with LLMs (Liu et al., 2025; Team, 2025; Wu et al., 2024; Chen et al., 2025). In this paper, we propose a novel image generation method that achieves better generation quality while enabling efficient sampling, promising to become a new paradigm for multimodal unified models.

### A.2  Hierarchical Planning for Long-horizon Manipulation

In long-sequence embodied operations, directly implementing end-to-end actions can lead to error accumulation due to a lack of intermediate guidance (Shridhar et al., 2022; Jiang et al., 2022; Nair et al., 2022). Therefore, many studies have adopted hierarchical planning, breaking down complex instructions into sequential subtasks for execution. Language planning utilizes large language models (LLMs) to transform reasoning into sequentially interpretable instructions in natural language (Ahn et al., 2022; Mu et al., 2023b). Some work also introduces explicit Chain of Thought (CoT) processes before generating actions (Zhang et al., 2025b; Zhao et al., 2025). Visual planning decomposes complex instructions into sequential subgoal images (Black et al., 2023b; Ni et al., 2024a), with additional efforts employing video generation models for task planning (Du et al., 2023; Soni et al., 2025). Multimodal planning integrates both approaches, generating interpretable instructions and subgoal images simultaneously, thereby providing rich planning information (Ni et al., 2024b; Zhang et al., 2025a).

### A.3  Reinforcement Learning in LLM and Image Generation

Reinforcement learning (RL) (Sutton & Barto, 1998) provides gradient-based policy optimization for non-differentiable learning objectives such as human preference alignment (Bai et al., 2022). However, applying RL to large language models (LLMs) remains challenged by inefficient sampling (Hu et al., 2024) and training instability (Zheng et al., 2023), driving substantial research efforts to develop robust policy gradient algorithms (Shao et al., 2024; Yu et al., 2025; Chu et al., 2025). While RL applications in image generation (Black et al., 2023a; Fan et al., 2023) have demonstrated partial success, existing approaches suffer from prohibitive computational costs due to iterative sampling requirements of autoregressive architectures, coupled with persistent convergence challenges. In this work, we address these limitations through an efficient sampling framework for unified multimodal generation, integrating RL with supervised learning objectives to enable reward maximization under strict maximum likelihood constraints.

## B  More Results and Analysis of Additional Experiments

### B.1  Other cases of RSFT

In this section, we conducted additional experiments with RSFT, referencing the experiments in DDPO (Black et al., 2023a). We designed two simple reward functions: compressibility and incompressibility, and implemented them using the same computation method as in DDPO. In practice, we performed the SFT task using an expert dataset and employed these two rewards to encourage the model to optimize for compressibility (or incompressibility). We present the reward curves(Figure 7) from the training along with the generation results.

The experimental curves demonstrate RSFT's robust convergence under both reward objectives, with visualization analysis revealing reward-specific generation patterns: images synthesized under the incompressibility objective exhibit sharpened features and intricate background complexity, while the compressibility-driven objective produces simplified outputs with attenuated background details. Crucially, these divergent behaviors remain anchored to coherent image structures through the maximum likelihood constraint – a core mechanism preventing arbitrary deviation from fundamental visual semantics. This effect originates from the method's dual optimization dynamics: reward gradients steer style adaptation while the likelihood penalty preserves content fidelity, achieving stable exploration within semantically grounded manifolds regardless of reward surface geometry.

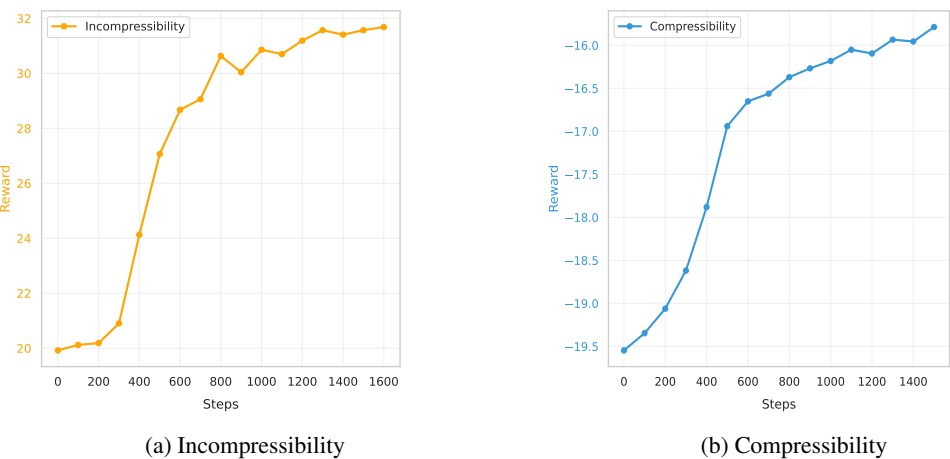

(a) Incompressibility
(b) Compressibility

Figure 7: RSFT training curves with Incompressibility (a) and Compressibility (b) as reward functions.

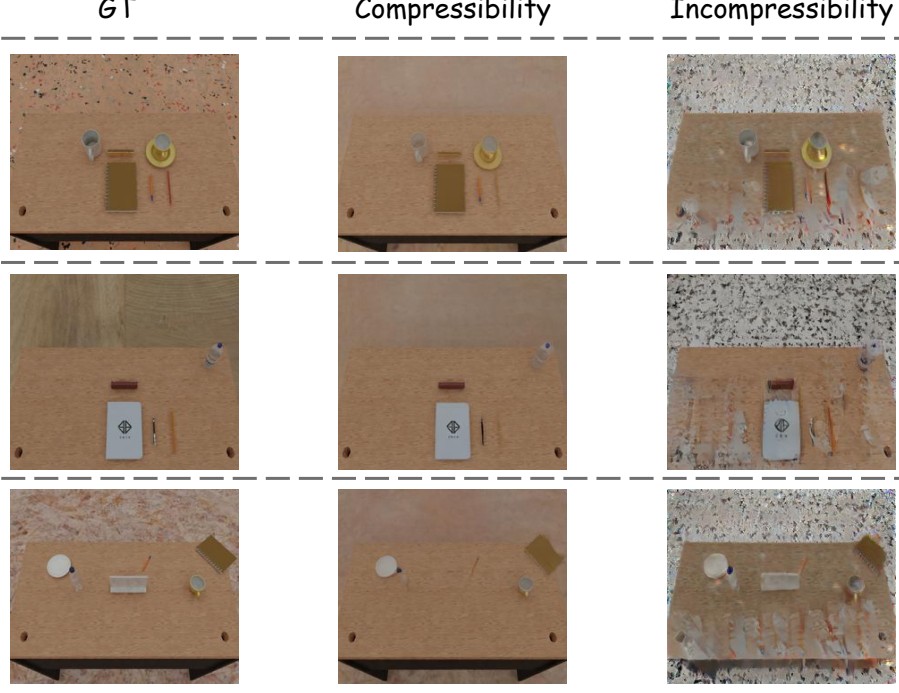

Figure 8: The generation effects on other rewards.

### B.2 VISUALIZATION OF DIFFERENT VARIANTS

Our ablation study reveals critical insights through visual comparisons of four variants: EVLP, EVLP-AR, EVLP-w/o-Se, and EVLP-w/o-En. EVLP achieves superior synthesis quality with precise semantic alignment and geometrically consistent layouts, whereas EVLP-AR suffers severe performance degradation due to error accumulation in autoregressive decoding – each iterative prediction step propagates spatial distortions that compound across generation steps. While EVLP-w/o-Se maintains basic object recognition, it fails to preserve fine-grained part relationships, producing anatomically implausible structures. Similarly, EVLP-w/o-En generates blurred textures with inconsistent lighting patterns, demonstrating the quantizer's role in disentangling high-frequency details from latent representations. These results quantitatively validate our architectural decisions through controlled component deactivation.

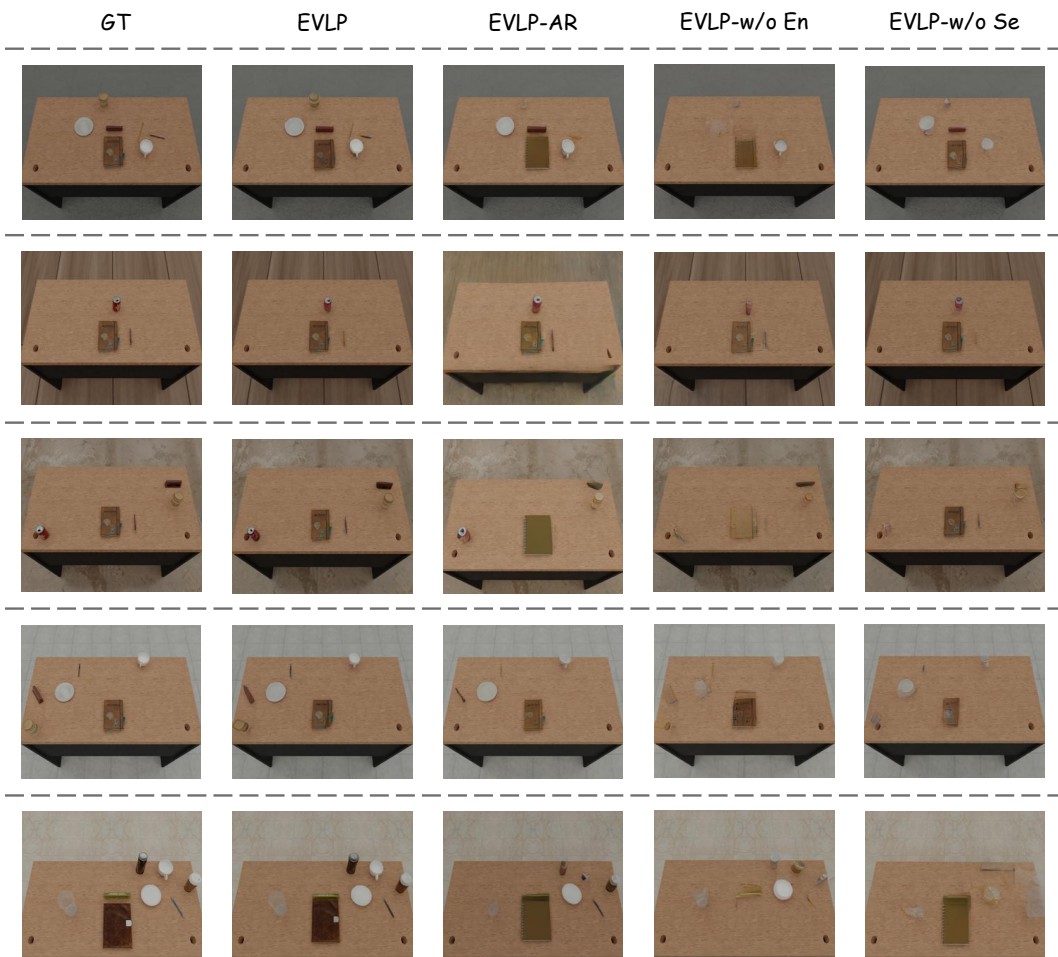

Figure 9: The generation effects of different variants.

### B.3 ABLATION ON MULTIMODAL PLANNING

In this section, we designed experiments to explore the advantages of multimodal tasks compared to unimodal tasks. Specifically, we conducted the EVLP w/o Gen experiment, in which the image generation task was removed from all stages, transforming the model into a purely language-planning system. We compared the success rates and language accuracy (Exps. A and B in Table 5). The results clearly demonstrate that multimodal tasks not only enhance the over-

Table 5: Comparative analysis of task planning performance across different variants.

| | Method | SR ↑ | LA ↑ |
|---|---|---|---|
| A | EVLP | 67.6 | 87.0 |
| B | - w/o Gen | 53.7 | 80.1 |

all success rate but also significantly improve language accuracy,
underscoring the positive impact of multimodal tasks on unimodal performance.

### B.4 SAMPLING EFFICIENCY

EVLP's single-step generative design can draw multiple *independent* samples in a single forward
pass, dramatically increasing sampling throughput compared with diffusion and autoregressive (AR)
models. Table 6 reports wall-clock generation time for producing 1 vs. 8 images. EVLP is orders of
magnitude faster in both 1.5B and 7B scales, and its advantage *widens* as model size grows.

Table 6: Image sampling time (seconds) for generating 1 and 8 images. Diffusion corresponds to
SuSIE; AR refers to autoregressive baselines. EVLP uses a single-step generator.

| Method | 1 Image (s) | 8 Images (s) |
|---|---|---|
| Diffusion (SuSIE) | 4.41 | 35.36 |
| AR-1.5B | 5.31 | 42.64 |
| AR-7B | 21.37 | 172.96 |
| **EVLP-1.5B** | **0.05** | **0.13** |
| **EVLP-7B** | **0.15** | **0.40** |

As shown, EVLP outpaces diffusion and AR by large margins (e.g., 0.15s vs. 21.37s for 7B, single
image), and the gap further enlarges when generating batches (8 images: 0.40s vs. 172.96s).

## C DETAILS OF METHOD

### C.1 REWARD FUNCTION

This reward function employs a dual-constraint mechanism to guide generative models in achieving
both physical dynamic consistency and visual coherence. The core design principles are: (1) Ensuring
spatiotemporal alignment of object motion trajectories between generated and real images; (2)
Enforcing that appearance changes during object motion adhere to physical interaction principles.
The implementation comprises three critical phases:

**Dynamic Region Detection**: The pipeline processes input image pairs $(x_t, x_{t+1})$ through Gaussian
blurring to enhance robustness against imaging noise, followed by differential computation and
morphological closing operations to extract significant change regions. Adaptive thresholding
combined with contour analysis identifies candidate bounding boxes, with non-maximum suppression
eliminating overlapping regions to produce final dynamic region set $\mathcal{B}$.

**Joint Feature Matching**: For ground truth and generated dynamic regions ($\mathcal{B}_{label}$ and $\mathcal{B}_{gen}$ respectively), we construct an IoU similarity matrix and apply the Hungarian algorithm for optimal bipartite
matching. Valid matches undergo pixel-level discrepancy analysis:

$$d_{ij} = -\text{MSE}(c_{ij}^{label}, c_{ij}^{gen}) \tag{6}$$

where $c_{ij}$ denotes normalized image patches.

**Multimodal Reward Computation**: Integrating spatial alignment and pixel-wise consistency, we
formulate the composite reward function:

$$r = \frac{\sum_{(i,j)\in\mathcal{M}}(\text{IoU}_{ij} + \lambda d_{ij}) - \tau(|\mathcal{B}_{label}| + |\mathcal{B}_{gen}| - 2|\mathcal{M}|)}{\min(|\mathcal{B}_{label}|, |\mathcal{B}_{gen}|)} \tag{7}$$

---

**Algorithm 1:** Dynamic-Aware Reward Computation

---

**Input:**
- Current state image $x_t$
- Generated image $x_{t+1}^{\text{gen}}$
- Real image $x_{t+1}^{\text{real}}$
- Hyperparameters: IoU threshold $\tau$, MSE weight $\lambda$, penalty coefficient $\gamma$

**Output:** Reward $r$

**Initialize dynamic regions**;

$\mathcal{B}_{\text{label}} = \text{DetectRegions}(x_t, x_{t+1}^{\text{real}})$ ;

$\mathcal{B}_{\text{gen}} = \text{DetectRegions}(x_t, x_{t+1}^{\text{gen}})$ ;

**Compute region matching**;

$$\mathbf{M} = \text{PairwiseIoU}(\mathcal{B}_{\text{label}}, \mathcal{B}_{\text{gen}})$$

$(row\_ind, col\_ind) = \text{Hungarian}(-\mathbf{M})$;

$\mathcal{M} = \{(i,j)|\mathbf{M}[i,j] \geq \tau\}$ ;

**Calculate multi-scale reward**;

$$score = \sum_{(i,j)\in\mathcal{M}} \left[\mathbf{M}[i,j] - \lambda \cdot \text{MSE}(c_{ij}^{\text{label}}, c_{ij}^{\text{gen}})\right]$$

where $c_{ij}^{\text{label}} = \text{Crop}(x_{t+1}^{\text{real}}, \mathcal{B}_{\text{label}}^i)$, $c_{ij}^{\text{gen}} = \text{Crop}(x_{t+1}^{\text{gen}}, \mathcal{B}_{\text{gen}}^j)$

**Apply penalty normalization**;

$$r = \frac{score - \gamma(|\mathcal{B}_{\text{label}}| + |\mathcal{B}_{\text{gen}}| - 2|\mathcal{M}|)}{\max(1, \min(|\mathcal{B}_{\text{label}}|, |\mathcal{B}_{\text{gen}}|))}$$

**return** $r$

---

## C.2 DETAILS OF PRETRAINING

During the dynamic perception pretraining phase, EVLP focuses on developing foundational multimodal generation capabilities through inverse and forward dynamic reasoning while enhancing spatial comprehension, change detection, and dynamic imagination. For generation, we employ Open-MAGVIT2-f16-262144 [1] as our image tokenizer, utilizing pretrained weights fine-tuned on our dataset for one epoch. The understanding module integrates SigLIP [2] as the semantic encoder and leverages the tokenizer's encoder for spatial encoding, with a convolutional pooling adapter (stride=2) bridging feature hierarchies. The architecture adopts Qwen2.5-1.5B-Instruct [3] as the LLM backbone, augmented with an auxiliary image head to project features into pixel probability distributions.

Throughout pretraining, SigLIP and tokenizer weights remain frozen while fully training other components. We implement a 4,000-step training regimen (16 hours) with batch size 2048, employing the AdamW optimizer (Loshchilov & Hutter, 2017) configured with learning rate 1e-4, 3% warmup proportion, and 0.01 weight decay. This configuration balances computational efficiency with gradient stability during large-scale multimodal alignment.

## C.3 DETAILS OF RSFT

We introduce in the main text a novel optimization algorithm termed Reinforced Supervised Fine-Tuning (RSFT), which effectively combines policy gradients with supervised fine-tuning (detailed implementation procedures are provided in Algorithm 2). In practical implementation, following pre-training completion, we initially conduct conventional supervised fine-tuning (SFT) for 1,000 steps with a batch size of 1,024, subsequently proceeding with 4,000 steps of RSFT training using the AdamW optimizer configured with a learning rate of 1e-4, 3% warmup proportion, and 0.01 weight decay - a configuration that optimally balances computational efficiency and gradient stability during large-scale multimodal alignment tasks.

---

[1] https://github.com/TencentARC/SEED-Voken/blob/main/docs/Open-MAGVIT2.md

[2] https://huggingface.co/google/siglip-base-patch16-256

[3] https://huggingface.co/Qwen/Qwen2.5-1.5B-Instruct

---

**Algorithm 2:** Reinforced Supervised Fine-Tuning (RSFT)

---

**Input:**
- Training dataset $\mathcal{D} = \{(g, x_t, a_t, x_{t+1})\}$
- Pretrained model $P_\theta$ with parameters $\theta$
- Reward function $R(\cdot)$
- Hyperparameters: batch size $B$, learning rate $\eta$, reward samples $K$, loss weight $\lambda$

**Output:** Optimized model parameters $\theta^*$

Initialize optimizer (e.g., Adam) with learning rate $\eta$;

**while** *not converged* **do**

    Sample a batch $\{(g^{(b)}, x_t^{(b)}, a_t^{(b)}, x_{t+1}^{(b)})\}_{b=1}^B$ from $\mathcal{D}$;

    **Compute SFT Loss $\mathcal{L}_{\mathbf{SFT}}$:**

$$\mathcal{L}_{\text{SFT}} = -\frac{1}{BL} \sum_{b=1}^B \sum_{i=1}^L \log P(a_t^{(i)}|a_t^{(<i)}, g, x_t) - \log P(x_{t+1}|g, x_t, a_t)$$

    **Compute Reinforcement Loss $\mathcal{L}_{\mathbf{RL}}$: for** *each sample* $b \in [1, B]$ **do**

        Generate $K$ image samples: $\{x_{t+1}^k\}_{k=1}^K \sim P(x_{t+1}|g^{(b)}, x_t^{(b)}, a_t^{(b)})$;

        Compute rewards: $r_k = R(x_{t+1}^k)$;

        Normalize rewards per batch: $\tilde{r}_k = \frac{r_k - \mu_r}{\sigma_r}$ where $\mu_r, \sigma_r$ are batch mean/std;

        Compute advantage: $A_k = \tilde{r}_k$;

    Compute policy gradient loss:

$$\mathcal{L}_{\text{RL}} = -\frac{1}{BK} \sum_{b=1}^B \sum_{k=1}^K A_k \cdot \log P(x_{t+1}^k|g^{(b)}, x_t^{(b)}, a_t^{(b)})$$

    **Joint Optimization:** Total loss: $\mathcal{L} = \mathcal{L}_{\text{SFT}} + \lambda \mathcal{L}_{\text{RL}}$;

    Update $\theta \leftarrow \theta - \eta \nabla_\theta \mathcal{L}$;

---

We further contextualize our algorithm against established RL baselines, specifically analyzing GPG (Chu et al., 2025) and the online variant of GRPO (Shao et al., 2024). The vanilla policy gradient method (GPG) directly optimizes action probabilities based on advantage estimates, fundamentally lacking stabilization mechanisms against high-variance gradient updates. GRPO-onpolicy addresses this instability through KL-divergence constraints between the optimized policy and a reference model – the latter pretrained on expert data via maximum likelihood estimation. While this prevents policy collapse, it introduces substantial computational overhead from maintaining dual models and propagating gradients through KL-divergence estimation. In contrast, RSFT circumvents these intermediate steps by directly incorporating expert supervision through maximum likelihood constraints on demonstration actions $\bar{a}$, seamlessly integrated into the policy gradient objective. This design eliminates the memory/computational bottlenecks of KL constraints while preserving expert alignment, as the additive log-probability terms explicitly regularize policy updates without reference model dependency. The absence of auxiliary models makes RSFT particularly advantageous in deployment scenarios requiring frequent policy updates or cross-domain adaptation, where GRPO-online's dependence on pretrained reference models becomes prohibitive due to potential expert data fitting errors and cascading approximation errors in KL estimation.

| RL Method | Loss Function | Constraint |
|---|---|---|
| GPG (Chu et al., 2025) | $\mathcal{L} = -\mathbb{E}\left[\frac{1}{K}\sum_{k=1}^K A_k \cdot \log \pi_\theta(a \mid s)\right]$ | No Constraint |
| GRPO-onpolicy (Shao et al., 2024) | $\mathcal{L} = -\mathbb{E}\left[\frac{1}{K}\sum_{k=1}^K A_k \cdot \log \pi_\theta(a \mid s) - \lambda D_{\text{KL}}(\pi_\theta(\cdot|s) \parallel \pi_{ref}(\cdot|s))\right]$ | KL Divergence |
| RSFT(Ours) | $\mathcal{L} = -\mathbb{E}\left[\frac{1}{K}\sum_{k=1}^K A_k \cdot \log \pi_\theta(a \mid s) + \lambda \log \pi_\theta(\bar{a} \mid s)\right]$ | Maximum Likelihood |

Table 7: Comparison of some RL methods.

# D DETAILS OF BENCHMARKS AND TASKS

**LoHoRavens**  LoHoRavens (Zhang et al., 2023) is a benchmark dataset based on the Ravens robot simulator, containing various long-horizon manipulation tasks. We categorize these tasks into three types: *Stacks*, *Sort*, and *Matching*. In *Stacks* tasks, the goal is to place blocks in absolute or relative positions; *Sort* tasks require sorting blocks or bowls with similar attributes; while *Matching* tasks involve attribute matching, such as placing blocks of corresponding colors into matching bowls. These tasks encompass multiple aspects of long-horizon reasoning, including color, size, space, arithmetic, and reference. To successfully complete each task, the robot must effectively integrate various reasoning capabilities and develop an appropriate long-term plan. In addition, we have extended a series of Letters tasks, which require the agent to understand information such as the shape and position of objects. We designed three types of tasks within this category: *Shape*, *Orders*, and *Spell*. Detailed task descriptions can be found in Table 8.

**Meeting Preparation**  Meeting Preparation is an in-house universal operational benchmark designed for real-world office scenarios, challenging agents to accomplish conference preparation tasks in highly diversified desktop environments. The test environment incorporates diverse office supply variants, with key item categories (e.g., cups) exhibiting rich morphological variations (mugs, glasses, vacuum flasks, etc.), combined with multiple environmental variables: desktop backgrounds of different materials, multiple camera perspectives, and dynamic object layouts. The core task demands agents to transcend superficial feature recognition and establish a semantic conceptualization framework—for instance, abstracting the common characteristics of "cups" from objects with varying heights, transparencies, and handle configurations, then strategically positioning them in task-appropriate areas according to conference requirements. This benchmark rigorously evaluates agents' capabilities in cross-domain generalization, fine-grained object recognition, and contextual reasoning, presenting critical challenges including (1) maintaining categorical cognitive stability when object functionality decouples from morphology, (2) interpreting scene intentions amidst multimodal interference. Examples of both benchmarks can be found in the Figure 10.

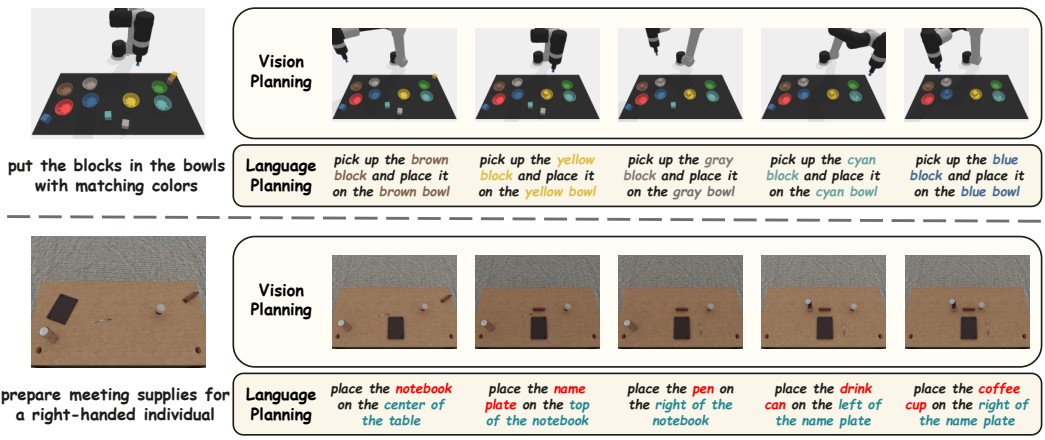

Figure 10: The example of benchmarks.

# E DETAILS OF DATASETS

For the LoHoRavens and Meeting Preparation datasets, we utilize the provided oracle engines to collect expert demonstration data. It is worth noting that if there are multiple correct answers or multiple ways to complete the task, we only focus on whether the instruction-specified complex task is ultimately accomplished and include all correct demonstrations as training data. For the 20 tasks in LoHoRavens, we collect 2,000 demonstrations per task, resulting in a total of 40,000 long-horizon demonstrations with sub-task counts ranging from 2 to 10+. In the Meeting Preparation environment, we gather 3,000 demonstrations. For all collected data, we split it into an 90% training dataset and a 10% testing dataset.

Table 8: Overview of three main task types: **Blocks**, **Letters**, and **Meeting**.

| Task Type | Description | Horizon | Color | Size | Spatial | Semantic |
|---|---|---|---|---|---|---|
| **Blocks** | | | | | | |
| Move | Move all the blocks to the [ABS POS] area | $4 \sim 15$ | ✗ | ✗ | ✓ | ✗ |
| | **Move all blocks of a color to the red zone** | $2 \sim 15$ | ✗ | ✗ | ✓ | ✗ |
| | Move all the blocks in the [ABS POS] area to the [ABS POS] area | $2 \sim 15$ | ✗ | ✗ | ✓ | ✗ |
| | **Move all the blocks on the corner/side** | $4 \sim 15$ | ✗ | ✗ | ✓ | ✗ |
| Stack | Stack all the blocks | $4 \sim 15$ | ✗ | ✗ | ✓ | ✗ |
| | **Stack blocks of the same size** | $4 \sim 15$ | ✗ | ✓ | ✓ | ✗ |
| | Stack blocks in alternate colors | $2 \sim 15$ | ✓ | ✗ | ✓ | ✗ |
| | **Stack only the primary color blocks on the left side** | $2 \sim 12$ | ✓ | ✗ | ✓ | ✗ |
| Matching | Put the blocks in the bowls with matching colors | $2 \sim 12$ | ✓ | ✗ | ✓ | ✗ |
| | **Put the blocks in the bowls with mismatching colors** | $2 \sim 12$ | ✓ | ✗ | ✓ | ✗ |
| | Put blocks of the same color in the zone with matching color | $2 \sim 12$ | ✓ | ✗ | ✓ | ✗ |
| **Letters** | | | | | | |
| Shape | Sort the vertically symmetrical letters to the bottom side | $2 \sim 15$ | ✗ | ✗ | ✓ | ✗ |
| | **Sort the horizontal symmetrical letters to the blank space** | $2 \sim 15$ | ✗ | ✗ | ✓ | ✗ |
| | Sort the central symmetrical letters to the corner | $2 \sim 15$ | ✗ | ✗ | ✓ | ✗ |
| Orders | **Put the letters on the tables in alphabetical order** | $2 \sim 15$ | ✗ | ✓ | ✓ | ✗ |
| | Put the letters on the tables in reverse alphabetical order | $2 \sim 15$ | ✗ | ✓ | ✓ | ✗ |
| | **Sort the consonants from all letters in orders** | $2 \sim 15$ | ✗ | ✓ | ✓ | ✗ |
| Spell | Spell words that are as long as possible | $4 \sim 15$ | ✓ | ✗ | ✓ | ✗ |
| | **Spell out the name of a top CS conference** | $4 \sim 10$ | ✓ | ✗ | ✓ | ✗ |
| | Spell out the name of a common transportation | $4 \sim 15$ | ✓ | ✗ | ✓ | ✗ |
| **Meeting** | | | | | | |
| Set up | Set up meeting supplies | $4 \sim 10$ | ✓ | ✓ | ✓ | ✓ |

During the dataset initialization process, we document each step's utilized assets and annotate their corresponding attributes, enabling precise identification of the color, size, and spatial relationships of the manipulated objects in each subtask's pick-and-place operations, thereby constructing accurate language actions. However, during testing, the model cannot access underlying environmental information, such as the exact count of real blocks or letters and their various attributes. The model must directly perceive from visual observations and reason based on these critical visual details, which significantly increases the task's difficulty.

## F    DETAILS OF BASELINES

**CLIPort**    CLIPort (Shridhar et al., 2022) is a popular end-to-end algorithm that functions as a language-conditioned imitation learning agent, directly processing high-level language instructions without requiring a planner. It integrates the broad semantic understanding of CLIP (Radford et al., 2021) with the spatial precision of Transporter (Zeng et al., 2021). As an end-to-end baseline, we employ CLIPort without modifications, training it by directly pairing high-level instructions with corresponding actions from the dataset.

**PAR**    PAR (Zhang et al., 2023) (Planner-Actor-Reporter) is a paradigm that replaces the skill predictor with a large language model (LLM) and employs a vision-language model (VLM) as a reporter for visual observations. Instructions and generated captions are then fed into the LLM for linguistic planning. In PAR, Llama 2 13B (Touvron et al., 2023) and the VLM OpenFlamingo (Awadalla et al., 2023) serve as the planner and reporter, respectively, using few-shot prompting. Notably, the Actor (or low-level foundational model) is precisely the language-conditioned CLIPort trained with step-by-step sub-instructions, as mentioned earlier. To ensure fair comparison, we make no modifications to it and keep the low-level foundational model consistent with CLIPort across all other baselines.

**EmbodiedGPT**    EmbodiedGPT (Mu et al., 2023b) represents a standard paradigm that integrates multimodal large language models (MLLMs) for language planning. The key distinction between

EmbodiedGPT and PAR lies in replacing the LLM+VLM combination with a more advanced MLLM, which exhibits superior visual reasoning capabilities. EmbodiedGPT leverages a constructed embodied chain-of-thought dataset to train the MLLM, enabling it to perceive visual details in hidden layers, similar to LLaVA (Liu et al., 2023b). To ensure fair comparison, we maintain consistency in the low-level foundational model across all baselines by using CLIPort as the common base.

**SuSIE**  SuSIE (Black et al., 2023b) introduces a hierarchical framework that employs an image-editing diffusion model as a high-level planner to generate intermediate subgoals for a low-level controller to execute. The method adopts InstructPix2Pix (Brooks et al., 2023) as its pre-trained image-editing model and fine-tunes it using language-annotated video clips and robot trajectory data from CALVIN (Mees et al., 2022). However, due to the sensitivity of image-editing models to training data, we observed limited generative performance in the Ravens domain. To address this, we conducted additional fine-tuning while strictly maintaining the same number of training iterations and dataset composition as in EVLP.

**CoTDiffusion**  CoTDiffusion (Ni et al., 2024a) represents a standard visual planning paradigm capable of translating complex instructions prompts—into visual subgoal images through a chain-of-thought reasoning process. The most notable distinction from SuSIE lies in CoTDiffusion's explicit incorporation of a semantic alignment module within its diffusion model. This module ensures correspondence and semantic coherence between generated images and high-level instructions, enabling chain-of-thought generation. Similar to SuSIE, we fine-tune CoTDiffusion on our collected dataset and employ the same low-level image-conditioned policy—specifically, the image-conditioned variant of CLIPort—for consistent implementation.

**PERIA**  PERIA (Ni et al., 2024b) represents a standard multimodal planning paradigm that integrates LLMs with image generation models to achieve unified multimodal planning. It supports the conversion of complex general instructions into multimodal sub-instructions (combining language actions with visual goals) while ensuring planning correctness through multimodal alignment. We fine-tune PERIA on our collected dataset and employ a multimodal conditioning strategy—specifically, the multimodal-conditioned variant of CLIPort—for implementation.

## G  DETAILS OF LOW-LEVEL POLICY LEARNING

To execute multimodal planning, we implement a low-level policy using CLIPort, chosen for its native SE(2) action space that is particularly suitable for Ravens benchmarks. We develop two variants: a language-conditioned policy trained with stepwise sub-instructions and an image-goal-conditioned policy trained with coherent keyframes, both serving as foundation models for planning tasks. For multimodal integration, we design a novel architecture that simultaneously processes image subgoals and language instructions through a 4-layer cross-attention network (4 heads, 768-dim embeddings). Training samples consist of expert trajectories $a$, observations $o$, instructions $e$, and subgoal images $v$ from $\mathcal{D}^{train}$. The policy $\psi$ minimizes the action prediction loss: $\mathcal{L}_{action} = \sum_{t=1}^{T} \|\hat{a}_t - p_\psi(a_t|o_t, e_t, x_t)\|_2$, optimized with AdamW (lr = 1e-4, 500-step warmup, weight decay 0.01) for $10^4$ steps (batch size 64). This approach provides dual advantages: (1) explicit subgoals reduce effective horizon complexity, and (2) combined language-visual conditioning enables segmented action prediction without requiring full-sequence conditioning on global instructions, significantly simplifying policy learning while maintaining execution quality.

## H  LIMITATION & FUTURE WORK

While EVLP demonstrates significant improvements in long-horizon manipulation tasks with complex instructions, several limitations warrant attention in future research:

- **Data Dependency**: The current implementation relies on pre-collected datasets for training the Multimodal Large Language Model (MLLM). While this approach effectively develops reasoning and planning capabilities, it may constrain the framework's adaptability to novel environments or tasks that substantially deviate from the training distribution. Future research could explore online

learning paradigms or adaptive fine-tuning methods to enhance EVLP's generalization capacity in unseen scenarios.

- **Sim-to-Real Gap**: Although EVLP exhibits strong performance in simulated environments, its effectiveness in real-world applications remains to be validated. Practical implementation introduces additional challenges including sensor noise, dynamic environmental changes, and physical interaction constraints that may impact system performance. Subsequent work should investigate EVLP's deployment on physical robotic platforms and rigorously evaluate its robustness and operational efficacy in authentic settings.

Despite these limitations, EVLP establishes a novel paradigm for robotic manipulation under general task instructions. By addressing these challenges through continued refinement, we anticipate that EVLP will inspire new research directions in long-horizon task execution with free-form instructions. This progression may ultimately contribute to developing more intelligent and versatile robotic systems capable of operating effectively across diverse real-world applications.

## I  SOCIAL IMPACT

The ALG framework enhances human-robot collaboration in manufacturing, healthcare, and domestic services by enabling natural instruction comprehension, boosting productivity and quality of life while creating new job opportunities. In education, ALG-powered robots guide children through adaptive learning activities like puzzle-solving, fostering cognitive development through personalized interactive tutoring. While advancing embodied AI capabilities, its deployment requires prioritized safety protocols, ethical compliance, and equitable accessibility to ensure responsible societal integration.

## J  LLM USAGE

Large language models (LLMs) were used solely to polish the writing (e.g., grammar correction and phrasing improvements). They did not contribute to research ideation.

