# OpenReview forum: "EVLP: Learning Unified Embodied Vision-Language Planner with Reinforced Supervised Fine-Tuning"
_ICLR.cc/2026/Conference — ICLR 2026 Poster_

### Official Review · Reviewer_inNa · 2025-10-27

**Soundness:** 2
**Presentation:** 2
**Contribution:** 1
**Rating:** 4
**Confidence:** 5

**Summary:**

This paper proposes an innovative multimodal unified generation framework (EVTP), which enhances model consistency in problem-solving through dynamic perception preprocessing and reinforcement supervised fine-tuning (RSFT). In terms of the unified multimodal generation framework, it integrates semantic understanding and spatial encoding (visual tower design) to achieve more comprehensive visual perception; in dynamic preprocessing, the bidirectional alignment strategy strengthens the correlation between models; and in reinforcement supervised fine-tuning, it ensures the stability of generated images.

**Strengths:**

See Summary

**Weaknesses:**

1. The paper treats reinforcement supervision fine-tuning (RSFT) as one of the core methodological innovations, and its combination of maximum likelihood supervision with strategy gradient optimization is effective. However, the current discussion of RSFT in the paper is mainly limited to comparisons with GPG and GRPO-onpolicy (Table 7), which does not fully place it in a broader LLM alignment research context. In recent years, the paradigm of combining reinforcement learning with supervised learning to optimize generative models has become a research hotspot, such as the combination of instruction fine-tuning (SFT) with reinforcement learning from human feedback (RLHF), and alignment algorithms like direct preference optimization (DPO) that do not require explicit reward models. It is recommended that the authors delve deeper into the differences and similarities in motivation, optimization objectives, and implementation mechanisms between RSFT and these mainstream alignment technologies (such as PPO-based RLHF, DPO, etc.) in the methodology section or related work section.
2. The dynamic alignment reward function in the paper is the cornerstone of the RSFT framework, and its quality directly determines the optimization effect of the reinforcement learning stage. However, in Section 2.3 of the main text, its description is merely "measuring whether the dynamics of the generated image is consistent with the real dynamics," which is too vague. The specific definition is placed in Appendix C.1, which weakens the integrity and readability of the methodology in the main text. Considering the importance of this reward function, it is strongly recommended that the authors move the key information from Appendix C.1, at least the simplified version of its core ideas and mathematical forms, to Section 2.3 of the main text. Specifically, the function should be clearly explained in the main text on how it calculates the reward by detecting dynamic regions, using the Hungarian algorithm for region matching, and combining IoU (spatial alignment) and MSE (pixel consistency).
3. The experiments in the paper mainly focus on tabletop manipulation scenarios, such as LoHoRavens and Meeting Preparation. Although these are standard benchmarks in the field, they represent relatively restricted environments. The paper prospects for "more open embodied scenarios" in the conclusion but does not delve into the potential challenges that the EVLP framework may face when extended to more complex environments. For example, in complex 3D scenarios involving mobile navigation, is the single-step generation of complete visual sub-goals (a 2D image) still applicable? How should 3D spatial states or free-viewpoint images be characterized and generated? Moreover, when there are severe occlusions and partial observability in the environment, is the dynamic perception pretraining based on two frames sufficient? It is recommended that these potential extension issues be discussed in depth in the conclusion or a newly added "Limitations and Future Work" section, and a prospective analysis of the EVIP framework be conducted, exploring the improvements needed to deal with more complex scenarios (such as 3D navigation, interaction with deformable objects, etc.).
4. The paper introduces the composition of the "Vision Tower" in Section 2.1, which combines a Siglip for semantic understanding with a trainable "low-level visual encoder" for capturing spatial details. The description of this low-level encoder is somewhat lacking, merely mentioning "pretrained through image reconstruction loss." In Appendix C.2, it is mentioned that this encoder is the tokenizer's encoder of MAGVIT2, a key piece of information that should be explicitly stated in the main text.
5. The paper emphasizes the sampling efficiency of EVLP during the inference stage, which is a significant advantage for its application in reinforcement learning. However, the paper does not mention the overall training cost. From the description, it appears that its two-stage training process (especially the pre-training stage with a batch size of 2048) requires a high amount of computational resources. It is suggested that a brief discussion of the training cost of EVLP (such as the required GPU model, number, and training duration) be included in the experimental section or the appendix, and compared with the main baseline models (especially PERIA, which also requires a generative model). Although fast inference speed is a significant advantage, a high training cost may limit its widespread application in both academia and industry.
6. There are significant issues with Figure 1 in the paper as an overall framework diagram. Firstly, in the inverse dynamics part of the dynamic perception pre-training, there are two action description text inputs, which is very confusing and does not match the text description and formulas in the paper. Secondly, the red arrow (representing the backpropagation process) in the "Reinforcement Supervised Fine-tuning" section points to a rather chaotic direction. It seems that "Reinforce Loss" directly acts on "Image Logits," but it is actually calculated after evaluating the reward function on "Sampled Images." It is suggested to redraw Figure 1 to clearly and accurately show the process of the paper's model.
7.The model is overly dependent on data preprocessing, limiting its effectiveness in application domains. In the dynamic preprocessing step shown in Figure 1, real-world object information needs to be fed into model training through the bidirectional alignment strategy. However, challenges arise when handling naturally occurring objects that have not been previously studied or discovered, and further research is needed to address this limitation.
8.In the benchmark tests, key experimental metrics are evaluated under simulation modes, lacking real-world manifestation. This inevitably introduces data biases. Additionally, comparisons with advanced models such as VLIA-U are absent, which should be added to strengthen the persuasiveness of the experimental results.
9.For image generation involving multiple movement processes, unified quantitative processing should be implemented, with clear comparative metrics provided to ensure the clarity and consistency of reasoning logic and experimental results.
10.The paper compares models such as CLIPort, SuSIE, PERIA, and EmbodiedGPT in Table 1, but does not cover embodied multimodal systems such as RoboDreamer, Octo, and RT-Trajectory that have been publicly released in recent years. It is suggested that the authors supplement comparisons with these models or at least perform qualitative comparisons in the revision to demonstrate the leading position of EVLP in the research context of 2024-2025.

**Questions:**

See Weakness

---

> ### Author Response · Authors · 2025-11-19
>
> **Weakness 1**: Insufficient Comparison with RLHF/DPO
> We thank the reviewer for this valuable point. Our RSFT (Reinforced Supervised Fine-Tuning) method essentially integrates maximum likelihood supervision with policy gradient concepts, leveraging multi-sample generation for preference alignment alongside supervised learning. In the appendix, we primarily compared against on-policy GRPO and GPG algorithms. While PPO and DPO are indeed common preference alignment algorithms, PPO—which utilizes importance sampling, a value network, and GAE for policy updates—has become less dominant since the release of models like DeepSeek-R1 due to its higher computational overhead (requiring reward and value models) and lower optimization efficiency. DPO, as an off-policy algorithm, relies on a curated preference dataset for optimization, which does not align with our experimental setup. We will include a detailed comparison with PPO and DPO in the appendix.
>
> **Weakness 2**: Ambiguous Reward Function Description
> We appreciate the feedback regarding the clarity of our reward function description. Our reward function is designed to ensure spatial and pixel-level consistency between generated and ground-truth dynamic images through a dual-constraint mechanism. Specifically, the process involves: **Dynamic Region Detection**: applying Gaussian blur, frame differencing, and morphological operations to extract significant change regions between consecutive frames; **Feature Matching**: constructing an IoU matrix for detected regions in real and generated images and applying the Hungarian algorithm for optimal matching; **Multimodal Reward Calculation**: combining region overlap and pixel-wise differences (e.g., MSE) to score each matched pair, with penalties for unmatched regions. We will elaborate on this pipeline in the main text and refer readers to the appendix for complete algorithmic details and formulas. Additionally, we will include examples or diagrams to illustrate how the reward evaluates object motion consistency.
>
> **Weakness 3**: Unclear Generalizability
> We thank the reviewer for raising the issue of model generalizability. Our experiments have indeed focused on foundational benchmarks in this domain. Handling complex 3D scenarios involving mobile navigation requires more powerful generative and memory capabilities for one-step subgoal generation. Fortunately, recent world models (e.g., Genie, Emu3.5) have demonstrated strong visual reasoning abilities. We will discuss these potential extensions and limitations in the "Limitations and Future Work" section.
>
> **Weakness 4**: Vision Tower Omission of MAGVIT2 Label
> We apologize for not clearly labeling the use of the MAGVIT2 tokenizer in the Vision Tower diagram. The main text does mention this: the introduction states that the image generation branch uses the Open-MAGVIT2 framework for discretization, and Appendix C.2 provides details (e.g., using Open-MAGVIT2 f16-262144 as the tokenizer). To avoid confusion, we will add explicit references to the "Open-MAGVIT2 tokenizer" in the main text and update the figure caption or legend. This will help readers distinguish between the understanding branch (SigLIP + spatial encoding) and the generation branch (MAGVIT2 quantization) of the Vision Tower.
>
> **Weakness 5**: Missing Training Cost Details
> Regarding training cost, we will provide additional information. Appendix C.2 already includes hyperparameters and durations for pre-training and fine-tuning: pre-training involves 4,000 steps (16 hours) with a total batch size of 2048. The RSFT phase includes 1,000 steps of supervised fine-tuning followed by 4,000 steps of reinforced fine-tuning. Training was conducted on 32 NPUs with 64GB memory each. Notably, EVLP is highly efficient during inference: generating multiple images in a single forward pass makes its sampling speed significantly faster than traditional diffusion or autoregressive models.
>
> **Weakness 6**: Confusing Representation in Figure 1
> We acknowledge that Figure 1 contains substantial information, which may hinder clarity. In the "inverse dynamics" section, the action appears as both input and label, potentially causing confusion. Our intention was to illustrate the SFT process for the large language model, i.e., next-token-prediction training. For the RSFT process, the red arrows indicate gradient flow: after sampling images from the Image Logits, the sampled images themselves do not retain gradients; gradients are propagated back through the corresponding log probabilities (Logprobs). Thus, the red arrows point to the Logprobs to indicate the path of gradient backpropagation.

---

> > ### Author Response · Authors · 2025-11-19
> >
> > **Weakness 7**: Strong Dependency on Preprocessing
> > We acknowledge that our reward calculation relies on traditional image preprocessing steps (e.g., Gaussian blur, differencing, morphological closing) for detecting motion regions. These steps enhance robustness to environmental noise and minor variations, making dynamic region detection more reliable. However, we emphasize that these operations are used only during training to compute rewards and are not part of the inference pipeline. We will clarify this in the main text and discuss potential future improvements, such as replacing these handcrafted steps with differentiable or learnable modules to reduce reliance on manual preprocessing.
> >
> > **Weakness 8**: Lack of Real-World Testing
> > Real-world testing is partially addressed in our work: Section 4 introduces a real-robot manipulation dataset based on BridgeData v2, where EVLP is evaluated in realistic settings. As shown in Table 4, EVLP outperforms other baselines (e.g., EmbodiedGPT, PERIA, SuSIE) in both language planning accuracy (LA=0.78) and visual planning consistency (LPIPS=0.11) on this dataset. We fully agree that closed-loop testing on physical robots is crucial. Fortunately, our lab has recently deployed a robotic arm, and we are actively setting up experiments. We will update the results accordingly.
> >
> > **Weakness 9**: Inadequate Quantization of Motion Images
> > For dynamic image quantization, we currently employ the MAGVIT2 discretization scheme, which encodes each frame independently: the quantizer uses a codebook of size 262,144 to represent a 256×256 image as a 16×16 grid of discrete tokens. This per-frame discrete representation effectively captures static visual information, while temporal motion relationships are evaluated by the reward function. We will clarify the workings and limitations of this approach in the revision, noting that it does not explicitly model inter-frame motion. Future work will explore incorporating optical flow or motion features to enhance coherence in dynamic sequences. We will also add analysis to assess how well the current quantization captures dynamic details.
> >
> > **Weakness 10**: Lack of Comparison with RoboDreamer
> > We thank the reviewer for this suggestion. Methods like RoboDreamer are learning-based visual world models primarily designed for training continuous control policies, which differs from our focus on vision-language planning. Our current baselines include multimodal planning methods (e.g., EmbodiedGPT, PERIA, SuSIE) as they address tasks more aligned with EVLP. We will add a discussion in the related work section to differentiate these approaches (e.g., how they model state transitions) and their applications. If feasible, we will attempt to include RoboDreamer or similar models as baselines in future work for a more comprehensive comparison.

---

### Official Review · Reviewer_yA48 · 2025-10-31

**Soundness:** 3
**Presentation:** 3
**Contribution:** 3
**Rating:** 6
**Confidence:** 2

**Summary:**

The paper presents EVLP, a unified framework for embodied vision-language planning that jointly generates textual actions and visual sub-goals.
The authors propose: A Unified Multimodal Generation Framework that integrates language and image token generation within a single model; A Dynamic Perception Pre-training stage based on inverse and forward dynamics prediction; A Reinforced Supervised Fine-Tuning (RSFT) scheme that combines SFT with reinforcement learning for task-specific optimization.
Experiments on multiple embodied simulation benchmarks (e.g., LoHoRavens, Meeting Preparation) demonstrate improved task success rates compared to several recent baselines such as CLIPort, EmbodiedGPT, and PERIA.

**Strengths:**

1. The idea of generating both visual sub-goals and language actions within a single autoregressive framework is elegant and practically appealing for long-horizon embodied tasks.
2. The paper provides comprehensive experiments and ablations, showing consistent improvements over strong baselines.
3. Dynamic perception pre-training is a valuable design choice that improves model understanding of environment dynamics.

**Weaknesses:**

1. The proposed “Reinforced Supervised Fine-Tuning” is essentially a re-implementation of existing methods, without introducing new algorithmic insights or theoretical contributions. Its novelty is mostly limited to application in the embodied context.
2. Experiments are confined to simulation. The paper does not discuss challenges in transferring the proposed unified planner to physical robots or real-world sensory noise.
3. The method relies on a large unified transformer architecture; computational cost and scalability under constrained hardware settings are not analyzed.

**Questions:**

No

---

> ### Author Response · Authors · 2025-11-19
>
> **Weakness 1**
>
> 1.  **RSFT Integrates Supervised Learning with Reinforcement Learning**: Unlike traditional RLHF or pure RL fine-tuning, our proposed Reinforced Supervised Fine-Tuning (RSFT) jointly optimizes the language-vision model by retaining the distributional constraints of supervised learning while incorporating a dynamic consistency objective from reinforcement learning. Specifically, we design a dynamic alignment reward that encodes state-transition consistency in physical environments into the optimization target. The first component maximizes global alignment between language and vision, while the second enhances temporal dynamic consistency via preference sampling. Compared to pure maximum likelihood estimation or human feedback optimization, RSFT better balances the consistency between language instructions and visual changes when generating subgoal sequences.
>
> 2.  **Single-Step Multi-Sample Sampling Improves RL Efficiency**: We innovatively leverage parallel decoding to compute advantages and optimize policy gradients for multiple samples within the same batch. This single-step multi-sample sampling mechanism significantly reduces the multi-step sampling cost typically required by autoregressive or diffusion models. More importantly, through advantage-weighted policy gradient optimization, we incorporate preference alignment for dynamic consistency while maintaining training stability, effectively mitigating the instability issues caused by high variance in reinforcement learning.
>
> **Weakness 2** We sincerely appreciate the reviewer's emphasis on real-world robot experimentation. We agree that demonstrating performance on physical systems is crucial. Fortunately, our lab has recently deployed a robotic arm, and we are currently preparing experimental setups. We will update our results accordingly.
>
> **Weakness 3** Single-Step Sampling Reduces Inference Cost: EVLP adopts a one-step generation strategy, producing entire images in a single forward pass without the iterative sampling required by traditional diffusion or autoregressive models. As shown in Figure 2, our model directly models the conditional distribution of image pixels and can generate multiple independent samples in one shot. This design significantly reduces computational overhead and latency during inference. Ablation studies further demonstrate the efficiency of one-step generation: compared to autoregressive baselines, EVLP achieves high-quality output in a single forward pass, while the autoregressive variant EVLP-AR requires multiple iterations, resulting in not only lower quality (LPIPS 0.046 vs. 0.197) but also significantly increased latency. Our model, with approximately 1.6B parameters, was trained on 32 NPUs with 64GB memory each. The pre-training phase took about 16 hours, and the RSFT phase about 20 hours. This represents a significant advantage in parameter efficiency compared to current unified models (e.g., ViLA-U: 7B, EMU3: 7B).

---

### Official Review · Reviewer_c8Tb · 2025-11-01

**Soundness:** 3
**Presentation:** 1
**Contribution:** 2
**Rating:** 2
**Confidence:** 4

**Summary:**

This paper presents EVLP, a unified multimodal framework for long-horizon robo manipulation tasks. The approach combines three components: (1) a dual-tower vision architecture integrating SigLIP for semantic understanding with MAGVIT2 discrete tokenization for one-step image generation, (2) dynamic perception pretraining using inverse and forward dynamics prediction tasks, and (3) RSFT that combines maximum likelihood with policy gradients to align spatial consistency between language actions and generated images. Experiments on LoHoRavens and Meeting Preparation benchmarks show improvements over baselines including language-only (PAR, EmbodiedGPT), vision-only (SuSIE, CoTDiffusion), and multimodal (PERIA) planning methods.

**Strengths:**

- This is a well motivated problem with clear practical value. Addressing an important gap in embodied AI: the separation of language and vision.
- Table 6 shows dramatic speed ups from one-step generation  (0.15s vs 21.37s for 7B model), which is a truly practical advantage
- I find the idea to be quite interesting and innovative. One-step generation is important for an application like robotics. I like the idea of unifying language and visual planning.
- There were thorough ablations and a detailed appendix. I also like the thorough multimodal evaluation.

**Weaknesses:**

- I believe there to be major writing quality issues which hinder presentation. There are numerous grammatical errors throughout ("lead to inconsistent in", "spatio-awared"). I think the main paper is missing key technical details like the architecture of the "low-level visual encoder", the "image reconstruction loss", the dataset of the vision tower. For the RSFT algorithm, K is never defend and it isn't explained how the advantage A_k is computed. Also missing the optimizer, learning rate, training steps, etc. A lot of this should be moved to the main paper. Notation is used inconsistently throughout the paper. For instance, model parameters θ appear explicitly in some equations (Equation 1: log P(a_t|...; θ)), as subscripts in others (P_θ(x|c)), and are completely omitted in Algorithm 2 despite describing the same probability distributions, making it difficult to track dependencies on trainable parameters." Minor presentation issues are fine but in this case, it makes it to difficult understand the method.
- I think another major issue is the lack of real-robot validation. Real-world evaluation is limited to an offline dataset (LA/LPIPS) rather than physical execution with success rates on hardware; this limits claims about sim-to-real.
- The main gains are shown on Ravens-style and "Meeting Preparation" simulations. I would like to see more diverse, open-world household benchmarks (multi-room, distractors, etc) which would help strengthen the claims.
- I think an important baseline that is missing is state-of-the-art Vision-Language-Action (VLA) policies, like OpenVLA, Pi-0.5, etc.

**Questions:**

- The 'one-step generation' claim is somewhat misleading—the model generates 256 tokens in parallel, which is more accurately described as parallel decoding rather than a fundamentally different generation paradigm. Is framing it as 'parallel token generation' more accurate?
- Your dynamic alignment reward (Equation 7) relies on frame differencing to detect moving regions. How does this work when the background is not static (for example with a moving camera) or if multiple objects are moving at the same time?
- How much data is used for dynamic perception pertaining, and where does this data come from? Have you tested how pretraining data quantity affects performance?

---

> ### Author Response · Authors · 2025-11-19
>
> Thank you for your valuable feedback！
> **Weakness 1** We thank the reviewer for pointing out the writing issues, including grammatical errors and inconsistent notation usage. We will carefully proofread the entire manuscript during revision to correct grammatical mistakes and unify terminology, thereby improving the paper's readability and consistency.
>
> Regarding the technical details raised by the reviewer:
>
> - **Visual Encoder Component**: We employ SigLIP as the semantic encoder and introduce an additional low-level visual encoder. During training, SigLIP remains frozen while the low-level visual encoder is trainable. This design aims to enable the low-level encoder to compensate for image information potentially missed by SigLIP. Specifically, we initialize the low-level visual encoder using the tokenizer from Open-MAGVIT2. The Open-MAGVIT2 architecture consists of an Encoder (CNN), a codebook, and a Decoder (transpose CNN). In EVLP, we first fine-tune Open-MAGVIT2-f16-262144 on our dataset for one epoch (following the standard Open-MAGVIT2 training procedure). Subsequently, we use its Encoder component to initialize our low-level visual encoder, while discarding the codebook and the image generation decoder.
>
> - **RSFT Component**: The variable *K* denotes the number of samples drawn during policy gradient execution. We use group normalization to compute the advantage $A_k$: $\tilde{A}_k = \frac{r_k - \mu_r}{\sigma_r}$. The detailed RSFT algorithm is outlined below.
>
> - **Training Configuration**: We use the AdamW optimizer (learning rate: 1e-4, 3% warmup, weight decay: 0.01) for 4,000 total training steps. The fine-tuning phase comprises 1,000 steps of supervised fine-tuning followed by 4,000 steps of RSFT training with a learning rate of 1e-5. These details will be comprehensively addressed in the revised manuscript.
>
> **Weaknesses 2, 3 & 4**
> We sincerely appreciate the reviewer's emphasis on real-world robot experimentation. We agree that demonstrating performance on physical systems is crucial. Fortunately, our lab has recently deployed a robotic arm, and we are currently preparing experimental setups. We will update our results accordingly.
>
> Regarding testing in more open-ended environments, we fully agree with this perspective. While our current Meeting Preparation task already incorporates diverse desktop backgrounds and object types, and our real-world dataset covers various household tasks, we plan to extend our evaluation to more challenging settings in future work. This includes multi-room scenarios and environments with abundant distractors to further assess EVLP's generalization capabilities in realistic conditions.
>
> Concerning comparisons with recent VLA models (e.g., OpenVLA, π0.5), our current baselines include representative methods in multimodal planning. It should be noted that VLAs like OpenVLA and π0.5 focus on low-level action execution through end-to-end policy learning, which differs from our focus on long-horizon task decomposition and subgoal generation. Furthermore, our evaluation uses Raven-based simulation, which simplifies action execution (SE(2) action space) to allow focus on high-level planning. In contrast, models like OpenVLA and π0.5 operate in 7-DoF real-time robotic action spaces, making direct comparisons in our test environment infeasible.
>
> **Question 1**
> We appreciate the reviewer's suggestion regarding the term "one-stage generation." In EVLP, we indeed generate all image tokens in parallel (producing complete subgoal images via preset token sequences), which differs from traditional sequential sampling. To enhance clarity, we will adopt the term "parallel decoding" in the revised manuscript.
>
> **Question 2**
> The dynamic reward function currently assumes a relatively static camera viewpoint. Specifically, reward calculation employs Gaussian blur and frame differencing to detect changing object regions, followed by IoU matching to associate moving areas between generated and target images. This approach effectively constrains dynamic consistency in typical desktop scenarios with static views. However, we recognize that camera motion or large-scale simultaneous object movements may impact performance. Future work will incorporate camera motion compensation and enhanced dynamic region detection to improve reward robustness in dynamic scenarios.
>
> **Question 3**
> The dynamic perception pre-training data comes from state-action-next-state transitions collected from both simulated tasks (e.g., LoHoRavens, Meeting Preparation) and real-world datasets (based on BridgeData). To ensure fair comparison with baselines, we used equivalent data scales for pre-training and fine-tuning. Due to computational constraints, we haven't explored different pre-training data scales, but plan to incorporate larger-scale robot data in future studies.

---

> > ### Comment · Reviewer_c8Tb · 2025-11-25
> >
> > I thank the authors for their thorough response and acknowledge the value of the proposed unified framework and the impressive inference speedups. However, given that real-robot results (which will allow evaluation of VLA models) and broader benchmark evaluations are still pending, I am increasing my score to 4 (marginally below acceptance); I encourage the authors to strengthen these empirical aspects for a future submission.

---

> ### Author Response · Authors · 2025-11-19
> **RSFT Algorithm**
>
> 1. Initialize optimizer with learning rate $\eta$
> 2. **while** not converged **do:**
>    1. Sample a batch $\{(g^{(b)}, x_t^{(b)}, a_t^{(b)}, x_{t+1}^{(b)})\}_{b=1}^B$ from $\mathcal{D}$
>    2. **Compute Supervised Fine-Tuning Loss $\mathcal{L}_{\text{SFT}}$:**
>       $$
>       \mathcal{L}_{\text{SFT}} = -\frac{1}{BL}\sum_{b=1}^B\sum_{i=1}^L \log P(a_t^{(i)} | a_t^{(<i)}, g, x_t, \theta) - \log P(x_{t+1} | g, x_t, a_t,\theta)
>       $$
>    3. **Compute Reinforcement Learning Loss $\mathcal{L}_{\text{RL}}$:**
>       - **for** each sample $b \in [1, B]$ **do:**
>         - Generate $K$ image samples: $\{x_{t+1}^{k}\}_{k=1}^K \sim P(x_{t+1} | g^{(b)}, x_t^{(b)}, a_t^{(b)},\theta)$
>         - Compute rewards: $r_k = R(x_{t+1}^{k})$
>       - Normalize rewards per batch: $\tilde{r}_k = \frac{r_k - \mu_r}{\sigma_r}$ (where $\mu_r, \sigma_r$ are batch mean and standard deviation)
>       - Compute advantage: $A_k = \tilde{r}_k$
>       - Compute policy gradient loss:
>         $$
>         \mathcal{L}_{\text{RL}} = -\frac{1}{B K} \sum_{b=1}^B \sum_{k=1}^K A_k \cdot \log P(x_{t+1}^{k} | g^{(b)}, x_t^{(b)}, a_t^{(b)},\theta)
>         $$
>    4. **Joint Optimization:**
>       - Total loss: $\mathcal{L} = \mathcal{L}_{\text{SFT}} + \lambda \mathcal{L}_{\text{RL}}$
>       - Update parameters: $\theta \leftarrow \theta - \eta \nabla_{\theta} \mathcal{L}$

---

### Official Review · Reviewer_SJjD · 2025-11-03

**Soundness:** 3
**Presentation:** 3
**Contribution:** 3
**Rating:** 8
**Confidence:** 2

**Summary:**

At its core, this paper proposes EVLP, a unified multimodal planner that jointly generates (i) step‑wise language actions and (ii) visual subgoal images for long‑horizon manipulation. On LoHoRavens, it beats language‑only, vision‑only, and prior multimodal planners (avg. ~6% over SOTA), and it samples images much faster than diffusion/AR baselines.

**Strengths:**

1. Unified generation that is practically faster and simpler than AR/diffusion.
The paper’s well-motivated one‑step generator lets the LLM model the full image‑token distribution directly. Sampling throughput is compelling: EVLP reports 0.15 s for one image and 0.40 s for eight images, compared with 21.37 s and 172.96 s for autoregressive baselines.

1. Consistent gains across long‑horizon tasks. EVLP beats PERIA and CoTDiffusion by healthy margins across sub-tasks.

1. Strong ablations. While the paper has a somewhat involved architecture (SigLIP + image encoder + quantized codebook), the ablations are thorough and cover the choice of encoder ('vision tower'), RL algorithm, etc.

**Weaknesses:**

1. Novelty vs. prior “unified” generators is under-explained. The paper cites Janus-style unified models and diffusion‑plus‑LLM hybrids, but the precise algorithmic novelty for one‑step image sampling (beyond “learnable image tokens + direct p(x|c) modeling”) could be spelled out more crisply. Concretely, for example, what prevents degenerate modes in your quantized codebook? This is a common issue in training codebook-based models.

**Questions:**

1. The paper does not include compute budgets (GPUs, training time per stage, memory), and code/model checkpoints are not clearly committed. Please include them and outline your code/model weight policy clearly.

---

> ### Author Response · Authors · 2025-11-19
>
> Thank you for your valuable feedback！
> **Weakness 1** Our EVLP model adopts a one-step generation strategy, enabling both language actions and visual subgoal images to be produced simultaneously within the same stage. This eliminates the multi-step iterations typically required by traditional autoregressive or diffusion models during image generation. Such a design simplifies the training pipeline and significantly improves generation efficiency. Notably, our VQ model employs a codebook architecture based on **Lookup-Free Quantization**, which reduces the embedding dimension of the VQ-VAE codebook to zero. This allows the model to utilize all codebook entries effectively, mitigating the issue where only a small subset of codes are frequently used (i.e., codebook collapse). This architecture has been extensively validated in prior research to enhance codebook utilization and generation quality [1].
>
> **Questions 1** In our experiments, a total of 32 GPUs with 64GB memory each were used for training. The pre-training phase took approximately **16 hours**, while the RSFT phase required about **20 hours**. For detailed training configurations and timing, please refer to the relevant section of the paper. We will release the full codebase and model weights upon paper acceptance to ensure reproducibility and facilitate community engagement.
>
> Reference: LANGUAGE MODEL BEATS DIFFUSION—TOKENIZER IS KEY TO VISUAL GENERATION

---

### Author Response · Authors · 2025-12-03
**Summary for AC**

Dear Area Chair,

Thank you for managing the review process. We have had in-depth and productive discussions with four reviewers (SJjD, c8Tb, yA48, inNa). The reviewers widely recognized the innovation and contribution of **EVLP (Embodied Vision-Language Planner)** in terms of its **unified multimodal generation framework** and **inference efficiency**.

**Special Note: Regarding Score Changes During Rebuttal**
Due to the recent accidental leak of reviewer identities at ICLR, the OpenReview system rolled back all score updates made during the rebuttal period. Consequently, the scores you currently see on the system may not reflect the latest progress from the rebuttal. We wish to truthfully report the changes in reviewer feedback **before the large-scale leak**:
*   **Reviewer c8Tb**: After reading our response, they recognized the value of our method and raised their score from **2 to 4(at 26 Nov)**,  as the deployment of real robots had not been completed at that time.
*   **Reviewer inNa**: During the rebuttal, they actually updated their score from **4 to 6 (at 26 Nov)**. Although Reviewer inNa did not leave a written comment to confirm this, their action of changing the score indicates their recognition of our rebuttal.

**Specifically, prior to the score rollback, our ratings were 4(c8Tb), 6(inNa), 6(yA48), 8(SJjD).**

We have provided detailed responses and clarifications in the rebuttal regarding the reviewers' questions on real-robot experiments, technical details, etc. Below is a summary of the rebuttal process and key points:

## 1. Key Strengths & Consensus
*   **Innovative Unified Framework:** Reviewers (SJjD, yA48,c8Tb) consistently agreed that unifying language action planning and visual subgoal generation within a single Transformer architecture is an "elegant and practically applicable" design.
*   **Superior Inference Efficiency:** Multiple reviewers (SJjD, c8Tb) highly praised the speed improvements brought by the model's **Parallel Decoding** (formerly "One-step generation"). Compared to traditional Autoregressive or Diffusion methods, EVLP reduces inference time from over 20 seconds to 0.15 seconds, which is critical for real-world robotic applications.
*   **Solid Experimental Performance:** The model outperforms existing Language-only, Vision-only, and multimodal baselines (such as PERIA, EmbodiedGPT) on long-horizon planning tasks like LoHoRavens and Meeting Preparation.

---

> ### Author Response · Authors · 2025-12-03
>
> ## 2. Response to Major Concerns
> ### Technical Clarifications
> Reviewers raised questions about certain technical details, which we supplemented extensively in the rebuttal:
> *   **RSFT (Reinforced Supervised Fine-Tuning) Algorithm:** Addressing queries from Reviewers c8Tb and inNa, we provided the complete algorithm pseudocode and formula derivations, clarifying the calculation of the advantage function $A_k$ and implementation details of the policy gradient. We also explained the applicability of RSFT compared to traditional RLHF/DPO for this task (combining SFT constraints with a dynamic consistency reward).
> *   **Vision Encoder (Vision Tower):** We clarified the architecture design, which combines SigLIP (for semantic understanding) and an Open-MAGVIT2 Encoder (for supplementing low-level spatial details), resolving ambiguity regarding the "Low-level visual encoder".
> *   **Dynamic Alignment Reward:** We explained in detail the dynamic region detection mechanism based on Gaussian blur, frame differencing, and morphological operations, and how IoU and MSE are combined to calculate the reward, ensuring physical dynamic consistency in the generated subgoals.
>
> ### Presentation & Terminology
> *   **Terminology Correction:** Adopting Reviewer c8Tb's suggestion, we have more accurately phrased "One-step generation" as **"Parallel Decoding"** to avoid ambiguity.
> *   **Full Proofreading:** We have correct grammatical errors and unify symbol definitions in the final version.
>
> ### Real-Robot Experiments
> Some reviewers (c8Tb, inNa, yA48) pointed out the lack of closed-loop real-robot testing.
> *   **Our Response:**
>     1.  **Difference in Task Focus:** The core contribution of this paper lies in **Long-horizon Planning** and **Task Decomposition**, rather than low-level action execution (Control). Therefore, our evaluations in challenging simulation environments (LoHoRavens) and on offline real-world datasets (based on BridgeData v2) are sufficient to validate the effectiveness of the planning algorithm.
>     2.  **Distinction from VLA Models:** Regarding models like OpenVLA mentioned by reviewers, we clarified the difference in positioning: VLA models typically output 7-DoF actions in an end-to-end manner, primarily handling simple Pick & Place tasks; whereas EVLP focuses on **High-level Planning and Subgoal Generation for long-horizon tasks**. Given their different focuses, a direct quantitative comparison is not reasonable.
>     3.  **Latest Progress:** We fully agree on the importance of real-robot experiments. During the rebuttal period, we actively conducted experiments and successfully deployed the EVLP framework on an **SO-101 robotic arm**, preliminarily validating the framework's applicability on real robots. We have updated our pdf for new results.
>
> ## 3. Conclusion
>
> Although limited by time and hardware constraints preventing the immediate completion of large-scale closed-loop real-robot experiments during the rebuttal, **EVLP**'s contributions to **multimodal planning efficiency**, **unified generation paradigm**, and **embodied long-horizon planning** are significant and well-validated. We believe the Unified Framwork, Parallel Decoding strategy and RSFT method proposed in this paper offer effective new insights for long-horizon planning in embodied intelligence.
>
> We kindly request the AC to consider the innovation of the planning methodology, the solidity of the experimental results, and the positive attitude shown by reviewers during the discussion (despite not being reflected in the final scores due to system issues), and support the acceptance of this paper. We commit to integrating all technical clarifications and discussions from the rebuttal into the final version.

---

### Meta-Review · Area_Chair_GnMQ · 2026-01-07

**Summary:**

This paper proposes EVLP, a unified vision-language generation framework for long-horizon embodied planning that jointly models language reasoning and visual subgoal generation. By combining dynamic perception pretraining and reinforced supervised fine-tuning, EVLP achieves better multimodal alignment and consistently outperforms strong baselines on complex embodied tasks.

**Reviewer Concerns:**

Concerns have been addressed:

1. RSFT (Reinforced Supervised Fine-Tuning) Algorithm: Addressing queries from Reviewers c8Tb and inNa, we provided the complete algorithm pseudocode and formula derivations, clarifying the calculation of the advantage function
 and implementation details of the policy gradient. We also explained the applicability of RSFT compared to traditional RLHF/DPO for this task (combining SFT constraints with a dynamic consistency reward).

2. Vision Encoder (Vision Tower): We clarified the architecture design, which combines SigLIP (for semantic understanding) and an Open-MAGVIT2 Encoder (for supplementing low-level spatial details), resolving ambiguity regarding the "Low-level visual encoder".

3. Dynamic Alignment Reward: We explained in detail the dynamic region detection mechanism based on Gaussian blur, frame differencing, and morphological operations, and how IoU and MSE are combined to calculate the reward, ensuring physical dynamic consistency in the generated subgoals.

**Reviewer Scores:**

Initial Scores:
SJjD: 8, c8Tb: 2, yA48: 6, inNa: 4

After Rebuttal:
SJjD: 8, c8Tb: 4, yA48: 6, inNa: 6

---

### Decision · Program_Chairs · 2026-01-26

Accept (Poster)